# Evaluation of Vascular Endothelial Function in Young and Middle-Aged Women with Respect to a History of Pregnancy, Pregnancy-Related Complications, Classical Cardiovascular Risk Factors, and Epigenetics

**DOI:** 10.3390/ijms21020430

**Published:** 2020-01-09

**Authors:** Ilona Hromadnikova, Katerina Kotlabova, Lenka Dvorakova, Ladislav Krofta

**Affiliations:** 1Department of Molecular Biology and Cell Pathology, Third Faculty of Medicine, Charles University, 10000 Prague, Czech Republic; katerina.kotlabova@lf3.cuni.cz (K.K.); lenka.dvorakova@lf3.cuni.cz (L.D.); 2Institute for the Care of the Mother and Child, Third Faculty of Medicine, Charles University, 14700 Prague, Czech Republic; ladislav.krofta@upmd.eu

**Keywords:** cardiovascular risk factors, vascular endothelial function, gestational hypertension, fetal growth restriction, preeclampsia, peripheral arterial tonometry, pregnancy-related complications, microRNA, screening, whole peripheral blood

## Abstract

The aim of the study was to examine the effect of previous pregnancies and classical cardiovascular risk factors on vascular endothelial function in a group of 264 young and middle-aged women 3 to 11 years postpartum. We examined microvascular functions by peripheral arterial tonometry and EndoPAT 2000 device with respect to a history of gestational hypertension, preeclampsia, fetal growth restriction, the severity of the disease with regard to the degree of clinical signs and delivery date. Besides, we compared Reactive Hyperemia Index (RHI) values and the prevalence of vascular endothelial dysfunction among the groups of women with normal and abnormal values of BMI, waist circumference, systolic and diastolic blood pressures, heart rate, total serum cholesterol levels, serum high-density lipoprotein cholesterol levels, serum low-density lipoprotein cholesterol levels, serum triglycerides levels, serum lipoprotein A levels, serum C-reactive protein levels, serum uric acid levels, and plasma homocysteine levels. Furthermore, we determined the effect of total number of pregnancies and total parity per woman, infertility and blood pressure treatment, presence of trombophilic gene mutations, current smoking of cigarettes, and current hormonal contraceptive use on the vascular endothelial function. We also examined the association between the vascular endothelial function and postpartum whole peripheral blood expression of microRNAs involved in pathogenesis of cardiovascular/cerebrovascular diseases (miR-1-3p, miR-16-5p, miR-17-5p, miR-20a-5p, miR-20b-5p, miR-21-5p, miR-23a-3p, miR-24-3p, miR-26a-5p, miR-29a-3p, miR-92a-3p, miR-100-5p, miR-103a-3p, miR-125b-5p, miR-126-3p, miR-130b-3p, miR-133a-3p, miR-143-3p, miR-145-5p, miR-146a-5p, miR-155-5p, miR-181a-5p, miR-195-5p, miR-199a-5p, miR-210-3p, miR-221-3p, miR-342-3p, miR-499a-5p, and miR-574-3p). A proportion of overweight women (17.94% and 20.59%) and women with central obesity (18.64% and 21.19%) had significantly lower RHI values at 10.0% false positive rate (FPR) both before and after adjustment of the data for the age of patients. At 10.0% FPR, a proportion of women with vascular endothelial dysfunction (RHI ≤ 1.67) was identified to have up-regulated expression profile of miR-1-3p (11.76%), miR-23a-3p (17.65%), and miR-499a-5p (18.82%) in whole peripheral blood. RHI values also negatively correlated with expression of miR-1-3p, miR-23a-3p, and miR-499a-5p in whole peripheral blood. Otherwise, no significant impact of other studied factors on vascular endothelial function was found. We suppose that screening of these particular microRNAs associated with vascular endothelial dysfunction may help to stratify a highly risky group of young and middle-aged women that would benefit from early implementation of primary prevention strategies. Nevertheless, it is obvious, that vascular endothelial dysfunction is just one out of multiple cardiovascular risk factors which has only a partial impact on abnormal expression of cardiovascular and cerebrovascular disease associated microRNAs in whole peripheral blood of young and middle-aged women.

## 1. Introduction

Pregnancy-related complications such as gestational hypertension (GH), preeclampsia (PE), and fetal growth restriction (FGR) are associated with the risk of later development of diabetes mellitus [1,2,3,4,5,6], metabolic syndrome [7,8], hypertension [3,4,5,6,9,10,11,12], kidney diseases [4], atherosclerosis [13,14], ischemic heart disease [4,9,10,15,16,17,18,19], myocardial infarcts [4,5,6,11,16,17], heart failure [4,5,6,11], stroke [4,5,6,9,10,11,15,16,17,19], and deep venous thrombosis [3,9,10]. 

Few studies have been performed to explore postpartum vascular changes in young and middle-aged women with respect to a comprehensive history of pregnancy [20,21,22,23]. The aim of the current study was to examine the effect of previous pregnancies (total number of pregnancies per woman, total parity per woman, and infertility treatment) on vascular endothelial function. In addition, we examined microvascular functions (peripheral endothelium) in women with a prior onset of pregnancy-related complications (gestational hypertension, preeclampsia, and fetal growth restriction), and compared them with controls, women with a history of normally progressing pregnancies, matched for age and time elapsed since delivery (3 to 11 years). 

We further studied the association between vascular endothelial function and the severity of the disease with respect to the degree of clinical signs and delivery date: PE without severe features (PE w/o SF), PE with severe features (PE w SF), PE without the presence of HELLP syndrome (hemolysis, elevated liver enzymes and low platelet count) (PE w/o HELLP), PE with the presence of HELLP syndrome (PE w HELLP), PE without the presence of FGR (PE w/o FGR), PE with the presence of FGR (PE w FGR), PE with the onset before 34 weeks of gestation (early PE), PE with the onset after 34 weeks of gestation (late PE), FGR with the onset before 32 weeks of gestation (early FGR), and FGR with the onset after 32 weeks of gestation (late FGR).

Besides, we studied the association between classical cardiovascular risk factors and the occurrence of vascular endothelial dysfunction. We compared Reactive Hyperemia Index (RHI) values and the prevalence of vascular endothelial dysfunction among the groups of women with normal and abnormal values of BMI, waist circumference, systolic (SBP) and diastolic (DBP) blood pressures, heart rate, total serum cholesterol levels, serum HDL (high-density lipoprotein) cholesterol levels, serum LDL (low-density lipoprotein) cholesterol levels, serum triglycerides levels, serum lipoprotein A Lp(a) levels, serum CRP (C-reactive protein) levels, serum uric acid levels, and plasma homocysteine levels. Furthermore, we determine the effect of the blood pressure treatment, the presence of trombophilic gene mutations, current smoking of cigarettes, and current hormonal contraceptive use on the vascular endothelial function.

The EndoPat test, the only FDA-cleared test for the non-invasive assessment of endothelial dysfunction (arterial health), clinically proven besides other things as a valuable tool in independent cardiovascular risk stratification for patients with early coronary athrosclerosis, was used [24,25].

This study is a follow-up of our previous study dedicated to the assessment and comparison of cardiovascular risk among women with normal and complicated course of gestation 3 to 11 years after the delivery [26]. In this context, women with a history of GH and PE were identified to be a risky group of patients that would benefit from implementation of early primary prevention strategies, since they had higher body mass index (BMI), waist circumferences, systolic (SBP) and diastolic (DBP) blood pressures, predicted 10-year cardiovascular event risk, and increased serum levels of uric acid and lipoprotein A, when compared with women with a history of normotensive term pregnancy [26]. We suggest that the extension of the study for early identification of the presence of endothelial dysfunction in young women, would be highly valuable, since endothelial dysfunction, which predicts cardiovascular events in early stages of disease, is still treatable and reversible, and usually occurs beyond any conventional risk factors. Some studies demonstrated in the group of young adult patients with type 1 diabetes (mean age 36 years) that vascular endothelial dysfunction occurred earlier than arterial stiffness [27]. 

Recently, we have also demonstrated that epigenetic changes characteristic for cardiovascular/cerebrovascular diseases are already present in whole peripheral blood of young and middle-aged women with a history of pregnancy-related complications and proposed a strategy to implement screening of cardiovascular/cerebrovascular disease associated microRNAs in primary prevention programmes to identify patients at a higher risk of later development of cardiovascular/cerebrovascular diseases as soon as possible [28].

In view of this fact, we also examined the association between the vascular endothelial function and postpartum expression of microRNAs involved in pathogenesis of cardiovascular/ cerebrovascular diseases (miR-1-3p, miR-16-5p, miR-17-5p, miR-20a-5p, miR-20b-5p, miR-21-5p, miR-23a-3p, miR-24-3p, miR-26a-5p, miR-29a-3p, miR-92a-3p, miR-100-5p, miR-103a-3p, miR-125b-5p, miR-126-3p, miR-130b-3p, miR-133a-3p, miR-143-3p, miR-145-5p, miR-146a-5p, miR-155-5p, miR-181a-5p, miR-195-5p, miR-199a-5p, miR-210-3p, miR-221-3p, miR-342-3p, miR-499a-5p, and miR-574-3p) [29,30,31,32,33,34,35,36,37,38,39,40,41,42,43,44,45,46,47,48,49].

## 2. Results

### 2.1. Impact of A History of Pregnancy and Pregnancy-Related Complications on vascular Endothelial Function in Young and Middle-Aged Women

Overall, no significant effect of total number of pregnancies per woman, total parity per woman, and infertility treatment on RHI values and the prevalence of endothelial dysfunction was observed (Table 1, Table 2 and Table 3). In parallel, no significant differences in RHI values and the prevalence of endothelial dysfunction was observed between the groups of young and middle-aged women with a history of pregnancy-related complications (GH, PE, and FGR) and normally progressing pregnancies (Table 4 and Table 5). Furthermore, statistical analyses showed no differences in RHI values and the prevalence of endothelial dysfunction between the control group and the groups of women with a history of PE w/o SF, PE w SF, PE w/o HELLP, PE w HELLP, PE w/o FGR, PE w FGR, early PE, late PE, early FGR, and late FGR (Table 4 and Table 5). Besides, the RHI values did not correlate with any of the studied pregnancy-associated factor (total number of pregnancies per woman, total parity per woman, time elapsed since the delivery) (Table 3).

### 2.2. Impact of Classical Cardiovascular Risk Factors on Vascular Endothelial Function in Young and Middle-Aged Women

In the studied group of young and middle-aged women (264 cases altogether) we did not observe any impact of the blood pressure treatment, the presence of trombophilic gene mutations, current smoking of cigarettes, current hormonal contraceptive use, SBP, and DBP on vascular endothelial function (Table 1, Table 2 and Table 3). Furthermore, no effect of serum levels of total cholesterol, HDL (high-density lipoprotein) cholesterol, LDL (low-density lipoprotein) cholesterol, triglycerides, lipoprotein A, CRP (C-reactive protein), uric acid, and plasma levels of homocysteine on vascular endothelial function was found (Table 1, Table 2 and Table 3).

Nevertheless, although the prevalence of endothelial dysfunction did not significantly differ between the groups of women with the presence and absence of central obesity, women with central obesity (waist circumference values ≥ 80 cm) showed significantly decreased RHI levels when compared with the group of women with normal values of waist circumference (waist circumference values < 80 cm) (unadjusted data: *p* = 0.006, data adjusted for the age of patients: *p* = 0.008), (Table 1 and Table 2).

In parallel, the ROC curve analyses revealed significantly lower RHI values in a substantial proportion of women with central obesity at 10.0% FPR both before and after adjustment of the data for the age of patients (Figure 1). 18.64% women had RHI values ≤1.442 before adjustment of the data for the age and 21.19% women had abnormal RHI values after adjustment of the data for the age at a specificity of 90.0% (Figure 1).

Similarly, while the difference in prevalence of endothelial dysfunction between the group with a normal BMI (BMI < 25) and the overweight group (BMI ≥ 25–<30) has not yet reached a statistical significance, significantly lower RHI values were found in overweight women compared to women with normal BMI (unadjusted data: *p* = 0.036, data adjusted for the age of patients: *p* = 0.039), (Table 1 and Table 2). The ROC curve analysis was able to identify abnormal RHI values (≤1.412) in 17.94% and 20.59% of overweight women before and after adjustment of the data for the age at 10.0% FPR (Figure 2).

### 2.3. Association between Higher Expression Rates of miR-1–3p, miR-23a-3p, and miR-499a-5p in Whole Peripheral Blood and the Occurrence of Vascular Endothelial Dysfunction in Young and Middle-Aged Women

Higher expression rates of miR-1-3p (*p* = 0.008), miR-23a-3p (*p* = 0.030), and miR-499a-5p (*p* = 0.010) were detected in whole peripheral blood of young and middle-aged women with the occurrence of vascular endothelial dysfunction (RHI ≤ 1.67, *n* = 86) when compared with those possessing normal vascular endothelial function (RHI > 1.67, *n* = 178), respectively (Table 6, Figure 3).

At 10.0% FPR, a proportion of young women with the occurrence of vascular endothelial dysfunction was identified to have up-regulated expression profile of miR-1-3p (AUC 0.0601, *p* = 0.005, sensitivity 11.76%), miR-23a-3p (AUC 0.583, *p* = 0.024, sensitivity 17.65%), and miR-499a-5p (AUC 0.598, *p* = 0.011, sensitivity 18.82%) (Figure 3). The dysregulation of these three microRNAs was mutually independent. All these 3 microRNAs were simultaneously dysregulated just in four patients with abnormal RHI values. MiR-1-3p dysregulation was followed by miR-23a-3p dysregulation just in 1 patient with abnormal RHI values. MiR-1-3p dysregulation was followed by miR-499a-5p dysregulation also just in 1 patient with abnormal RHI values. The various combinations of miR-1-3p, miR-23a-3p, and miR-499a-5p were not superior over using only individual microRNA biomarkers. In addition, the RHI values showed a week negative correlation with expression of miR-1-3p (*ρ* = −0.156, *p* = 0.012), miR-23a-3p (*ρ* = −0.154, *p* = 0.013), and miR-499a-5p (*ρ* = −0.193, *p* = 0.002) in whole peripheral blood of young and middle-aged women. That means that women with a finding of lower RHI values showed significantly increased levels of miR-1-3p, miR-23a-3p, and miR-499a-5p in whole peripheral blood (Table 7, Figure 4).

## 3. Discussion

The majority of studies on the assessment of vascular endothelial function using peripheral arterial tonometry and EndoPAT 2000 device was performed in women during gestation with the aim to describe potential association between vascular endothelial dysfunction and pregnancy-induced hypertension or preeclampsia [20,23,50,51,52,53], sleep-disordered breathing in women with preeclamptic toxaemia [54], and smoking status [55].

However, only few studies have been performed to explore postpartum vascular changes in young and middle-aged women with respect to a comprehensive history of pregnancy [20,21,22,23].

In general, our study indicated no significant effect of a history of pregnancy with respect to total number of pregnancies per woman, total parity per woman, infertility treatment, particular pregnancy-related complication subtypes (GH, PE, and FGR), the severity of the disease with regard to the degree of clinical signs (PE w/o SF, PE w SF, PE w/o HELLP, PE w HELLP, PE w/o FGR, and PE w FGR), and delivery date (early PE, late PE, early FGR, and late FGR) on vascular endothelial function. Neither RHI values or the prevalence of endothelial dysfunction differed between particular groups of young and middle-aged women. Parallel, no correlation between RHI values and total number of pregnancies per woman, total parity per woman, and time elapsed since the delivery was found.

These data are in agreement with the data of previous follow-up study accomplished 6–9 months postpartum [20], which observed no significant differences in RHI values between women after uncomplicated pregnancies and women who developed pregnancy-induced hypertension or preeclampsia.

The current study also produced similar findings to another follow-up study performed on young women with a history of preeclampsia from a semi-rural region of South Africa [23], where no significant differences in RHI values were found between the group of non-pregnant women, the preeclampsia group during gestation, and the group of women with a history of preeclampsia between subsequent visits after the delivery, which had been carried up from 15 weeks to 1 year postpartum.

Nevertheless, our data partially contradict the findings of other investigators [21,22], who observed that a history of PE was associated with lower levels of RHI. They demonstrated that women with previous pregnancy complicated by early PE or late PE showed a significant reduction in RHI values between 6 months and 4 years after delivery (mean time from delivery was around 2.3 years) when compared with controls [22]. However, like Orabona et al. [22] we found no significant differences in RHI values among PE groups: women who had experienced HELLP and those with a history of PE without HELLP. Concerning a history of FGR, our data are also consistent to a certain extent with the data resulting from the study of Orabona et al. [22], who described more frequent endothelial dysfunction in women, who experienced FGR that required the delivery before 34 weeks of gestation. Although the difference in prevalence of vascular endothelial dysfunction has not yet reached a statistical significance in our study, we observed a higher incidence of vascular endothelial dysfunction in women with a history of early FGR, whose pregnancy had to be terminated before 32 weeks of gestation, than in controls (66.67%vs. 32.43%, *p* = 0.057).

To clarify this matter, consecutive large scale studies are needed to examine the association between the previous occurrence of fetal growth restriction, especially with regard to the delivery date, and the vascular endothelial dysfunction. In addition, further studies, at the more advanced level, are needed to demonstrate if the appearance of vascular endothelial dysfunction is a consequence of a former onset of pregnancy-related complication or the other way around if the asymptomatic endothelial dysfunction has already appeared before gestation and might cause secondary placental insufficiency leading to fetal growth restriction. Besides, it is essential to focus the attention not only on the comparison of RHI values between particular groups, but to explore the prevalence of vascular endothelial dysfunction in individual groups as well.

Subsequently, we performed the association study with the aim to discover potential correlation between classical cardiovascular risk factors and the occurrence of vascular endothelial dysfunction. We studied the effect of multiple factors such as the blood pressure treatment, the presence of trombophilic gene mutations, BMI, waist circumference, SBP, DBP, heart rate, total serum cholesterol levels, serum HDL cholesterol levels, serum LDL cholesterol levels, serum triglycerides levels, serum Lp(a) levels, serum CRP levels, serum uric acid levels, and plasma homocysteine levels on the vascular endothelial function via comparison of RHI values and the prevalence of vascular endothelial dysfunction among the groups of women with normal and abnormal clinical findings. Furthermore, we evaluated the impact of current smoking of cigarettes and current hormonal contraceptive use on vascular endothelial function. No impact of studied factors on the occurrence of vascular endothelial dysfunction was found and the RHI values did not correlate with any factor as well. Nevertheless, significantly decreased RHI levels were detected in overweight women (BMI ≥ 25–<30) and simultaneously in a group of women with central obesity (waist circumference values ≥ 80 cm). A substantial proportion of overweight women (17.94% and 20.59%) and women with central obesity (18.64% and 21.19%) had significantly lower RHI values at 10.0% FPR both before and after adjustment of the data for the age of patients.

Our results support the data of Lobysheva et al. [56], who observed as well no difference in RHI values between subjects consuming combined contraceptives and control subjects not using contraceptive pills, and no effect of combined contraceptives consumation on RHI values in short-term (less than 1 year usage) and long term periods (more than 3 years of usage).

A serious of studies was performed with the aim to identify potential factors affecting the vascular endothelial function [24,25,57,58,59,60,61,62,63,64,65,66,67,68,69,70,71,72,73]. Nevertheless, there were substantial variations at study designs and the groups of patients, which were subjects of interest. Naturally, most studies was focused mainly on elderly patients. However, limited data concerning the association between classical cardiovascular risk factors and vascular endothelial function are available entirely in the group of healthy young and middle-aged women. Ferreira et al. [74] described negative correlation between RHI values and serum uric acid in healthy Brazilian young and middle-aged adults (age 20–55 years). Although the studied group involved both sexes, these data may be in compliance with our data obtained in the group of young and middle-aged women with a history of complicated and uncomplicated pregnancy, where a trend toward impaired endothelial function (lower RHI levels) was observed in patients with increased serum levels of uric acid (*ρ* = −0.111, *p* = 0.072).

Another study [75] examined an association between vascular endothelial function and similar parameters as our study (age, sex, BMI, waist circumference, resting heart rate, SBP, DBP, serum total cholesterol levels, serum HDL cholesterol levels, serum LDL cholesterol levels, serum triglycerides levels, serum CRP levels, the blood pressure treatment, and the smoking status) in the group of middle-aged Finnish municipal workers (mean age 44.6 years). No of the following factors (age, SBP, DBP, serum total cholesterol levels, serum LDL cholesterol levels, serum CRP levels, and the blood pressure treatment) was reported to affect RHI values in sex-adjusted models, which is in compliance with the data resulting from our study performed on the group of young and middle-aged women after complicated and uncomplicated pregnancy. Nevertheless, in a study of Konttinen et al. [75] lower RHI values were strongly associated in separate linear regression sex-adjusted models with BMI, waist circumference, serum HDL cholesterol levels, serum triglycerides levels, and current smoking of cigarettes. Konttinen et al. [75] finally created a multivariable sex-adjusted model that was able to explain 20.0% of the variability of RHI in women. This final model for women involved HDL cholesterol, BMI and smoking status. In compliance with a study of Konttinen et al. [75] our independent study found decreased RHI levels in a group of overweight women, and in women with central obesity. In contrast to a study of Konttinen et al. [75] we did not observe the influence of current smoking of cigarettes and serum HDL cholesterol levels on RHI values.

The study of Williams et al. [76] demonstrated that childhood obesity sustained into early midlelife (till the age of 38 years) was associated with chronic endothelial dysfunction that contributed to an increased risk of cardiovascular events. At the age of 38 years, after adjustment of the data for sex, endothelial dysfuntion was associated with higher BMI, higher waist circumference, low HDL cholesterol levels, low cardiorespiratory fitness, and increased CRP levels [76]. Although, we did not have any information on BMI trajectories from childhood till the beginning of pregnancy in a group of women we examined, and we could make just the analyses regardless of the fact if the women had been overweight or obese from childhood, our data partially confirmed the findings of Williams et al. [76] that lower RHI levels are more frequently present in young and middle-aged overweight women and women with central obesity with waist circumference values exceeding 80 cm. 

In addition, Randby et al. [77] showed sleep apnoe as an independent predictor of an impaired vascular endothelial function in middle-aged women (mean age 48 years). Similarly, the low serum 25-hydroxy vitamin D levels were significantly related to lower RHI values in healthy middle-aged (mean age 41 years) normotensive, non-smoker, normolipidemic and normoglycemic women [78]. In our study we did not focus on the examination of the presence of sleep apnoe or serum 25-hydroxy vitamin D levels, but we believe that the findings resulting from previous studies [77,78] might contribute to the explanation of low RHI values in a proportion of healthy young and middle-aged female population. But still a large part of a lowered RHI values remains accounted for unknown factors [75].

To our knowledge, no study examining association between expression profiles of cardiovascular/cerebrovascular disease associated microRNAs in whole peripheral blood of non-pregnant women and vascular endothelial function assessed by peripheral arterial tonometry and EndoPAT 2000 device had been carried out. Just the association between serum miR-126 levels and improvement of microvascular endothelial dysfunction induced by exercise and diet was described in obese adolescents [79]. That’s why we further studied the association between the vascular endothelial function and postpartum expression of microRNAs involved in pathogenesis of cardiovascular/cerebrovascular diseases (miR-1-3p, miR-16-5p, miR-17-5p, miR-20a-5p, miR-20b-5p, miR-21-5p, miR-23a-3p, miR-24-3p, miR-26a-5p, miR-29a-3p, miR-92a-3p, miR-100-5p, miR-103a-3p, miR-125b-5p, miR-126-3p, miR-130b-3p, miR-133a-3p, miR-143-3p, miR-145-5p, miR-146a-5p, miR-155-5p, miR-181a-5p, miR-195-5p, miR-199a-5p, miR-210-3p, miR-221-3p, miR-342-3p, miR-499a-5p, and miR-574-3p) [29,30,31,32,33,34,35,36,37,38,39,40,41,42,43,44,45,46,47,48,49]. The expression profiles of miR-1-3p, miR-23a-3p, and miR-499a-5p were different between young and middle-aged women with normal vascular endothelial function (RHI > 1.67) and vascular endothelial dysfunction (RHI ≤ 1.67), respectively. In a proportion of young and middle-aged women with vascular endothelial dysfunction (RHI ≤ 1.67) higher expression rates of miR-1-3p (11.76%), miR-23a-3p (17.65%), and miR-499a-5p (18.82%) were detected at 10.0% FPR. In addition, lower RHI values correlated with increased expression levels of miR-1-3p (*ρ* = −0.156, *p* = 0.012), miR-23a-3p *(ρ* = −0.154, *p* = 0.013), and miR-499a-5p (*ρ* = −0.193, *p* = 0.002) in whole peripheral blood of young and middle-aged women. We suppose that screening of these particular microRNAs associated with vascular endothelial dysfunction may help to stratify a highly risky group of young and middle-aged women that would benefit from early implementation of primary prevention strategies.

Nevertheless, overall, it is obvious, that vascular endothelial dysfunction is one out of multiple cardiovascular risk factors which has only a partial impact on abnormal expression of cardiovascular and cerebrovascular disease associated microRNAs in whole peripheral blood of young and middle-aged women. That´s the reason, why we have been continuously working on the identification of all cardiovascular risk factors causing abnormal expression of cardiovascular and cerebrovascular disease associated microRNAs in whole peripheral blood of young and middle-aged women.

MiR-1-3p is generated from miR-1-1 and miR-1-2 precursors whose genes are located on chromosomes 20q13.3 and 18q11.2. MiR-1 is highly expressed in heart muscle, especially in the myocardium, and in skeletal muscles [80,81,82]. Circulating miR-1 levels are significantly elevated in acute myocardial infarction and correlate with circulating troponin T levels, a marker of cardiac damage [83]. MiR-1 is a potential target of therapeutic intervention in cardiovascular diseases, cardiac ischemia and complications following myocardial infarction. Inhibition of miR-1 by oligonucleotides is cardioprotective, as it leads to a reduction in apoptosis, an increase in resistance to oxidative stress and a reduction in spontaneous arrhythmias [84,85,86]. In addition, SUR2B/Kir6.1 channel openers correct endothelial dysfunction in chronic heart failure via the miR-1-3p/ET-1 pathway [87]. We recently reported that a proportion of women with up-regulated postpartal miR-1-3p expression profile in whole peripheral venous blood, who had a history of gestational hypertension or preeclampsia in their previous pregnancies, had an increased cardiovascular risk and should be monitored long-term [28]. Parallel, up-regulated miR-1-3p profile was observed in placental tissue and umbilical cord blood samples in women with the onset of preeclampsia and FGR [88,89]. But miR-1-3p expression levels did not show any changes in whole peripheral blood of women during gestation when patients with and without pregnancy-related complications were compared [90].

MiR-23a, encoded by a gene located at chromosome 19p13.12, forms two mature microRNAs: miR-23a-5p and miR-23a-3p. MiR-23a regulates cardiomyocyte apoptosis, a key pathogenesis factor of heart failure, by targeting manganese superoxide dismutase gene [91]. MiR-23a also regulates the vasculogenesis of coronary artery disease via targeting epidermal growth factor receptor [92]. Circulating miR-23a may be a new biomarker for coronary artery disease, since increased levels of miR-23a predict the presence and severity of coronary lesions in patients with coronary artery disease [92]. MiR-23a-3p also suppressed oxidative stress injury in a mouse model of focal cerebral ischemia-reperfusion [93]. Furthermore, miR-23c, but not miR-23a was reported to function as a new regulator to inhibit angiogenesis by targeting SDF-1α, which is responsible for delayed process of wound healing in patients with diabetic foot ulcer due to reduced expression of growth factors, persistant inflammation and endothelial dysfunction [94]. Since our recent study demonstrated upregulation of miR-23a-3p in a proportion of children with normal clinical findings born of gestational hypertension complicated pregnancies only, we suppose that compensatory effect of miR-23a-3p may appear more likely in these children to normalise cardiomyocyte state and vasculogenesis [95]. No change in miR-23a-3p expression profile was observed in various biological samples (whole peripheral venous blood, placental tissue, umbilical cord blood) when women with pregnancy-related complications and normal pregnancies were compared both during gestation or postpartum periods [88,89,90].

MiR-499a-5p, encoded by the miR-499a gene located on chromosome 20q11.22, is strongly expressed in the heart under physiological conditions. MiR-499 is involved in inhibiting cardiomyocyte apoptosis by suppressing calcineurin-mediated dephosphorylation of the Drp1 protein. It has also been reported that p53 downregulates the expression of miR-499 [96]. MiR-499 is also associated significantly with myocardial infarction, and together with miR-133a and miR-208a, they represent potential early diagnostic biomarkers of myocardial infarction [97,98,99,100]. Furthermore, miR-499 is significantly involved in the inflammatory signaling pathways of bronchial asthma [101]. No data on the role of miR-499a-5p in pathogenesis of endothelial dysfunction are currently available. Our recent study demonstrated upregulation of miR-499a-5p in a proportion of women with a history of gestational hypertension in previous pregnancy. Considering the clinical relevance of miR-499, we concluded that the group of women with the upregulated miR-499a-5p expression profile represented a high-risk group of patients who would benefit from early implementation of prevention programs and long-term monitoring [28]. Placental tissue demonstrated up-regulated miR-499a-5p expression profile in women with the occurence of gestational hypertension, preeclampsia and FGR [88]. Nevertheless, in whole periheral venous blood of women with the onset of gestational hypertension miR-499a-5p levels were significantly decreased [90].

## 4. Materials and Methods

### 4.1. Participants

The prospective study run from 2016 to 2019 in the Institute for the Care of the Mother and Child, Prague, Czech Republic and Second Internal Clinics of Cardiology and Angiology, General Faculty Hospital, Prague, Czech Republic.

All Caucasian women who delivered in the Institute for the Care of the Mother and Child, Prague, Czech Republic within 2007–2013 and were diagnosed with gestational hypertension (GH), preeclampsia (PE) and/or fetal growth restriction (FGR) were invited to participate in the study. Finally, 264 women participated in the study (48 GH, 114 PE with or without FGR, 28 FGR, and 74 women with normal course of gestation. The recruitment rate was 27.10% (264 out of 974 invited patients). Women with normal course of gestation were chosen on the basis of equal age and time elapsed since the delivery.

Twenty-eight out of 114 patients had symptoms of preeclampsia without severe features (PE w/o SF) and 86 out of 114 patients were diagnosed with preeclampsia with severe features (PE w/SF). In 40 PE patients gestation was terminated before 34 weeks of gestation (early PE) and 74 patients delivered after 34 weeks of gestation (late PE). Sixteen out of 114 patients had PE with HELLP syndrome. Nine pregnancies with growth-retarded foetuses were terminated before 32 weeks of gestation (early FGR) and 19 FGR pregnancies delivered after 32 weeks of gestation (late FGR).

Women with a history of normal pregnancies had no medical, obstetrical, or surgical complications at the time of the delivery and delivered full term, singleton healthy infants weighing > 2500 g after 37 completed weeks of gestation.

Gestational hypertension was defined as high blood pressure that developed after the twentieth week of pregnancy [11,102,103,104].

Preeclampsia was defined as blood pressure > 140/90 mmHg in two determinations 4 h apart that was associated with proteinuria > 300 mg/24 h after 20 weeks of gestation in a woman with a previously normal blood pressure. In the absence of proteinuria, any of the following can establish the diagnosis of PE: (1) new-onset thrombocytopenia, (2) impaired liver function, (3) renal insufficiency, (4) pulmonary edema, or (5) visual or cerebral disturbances. Severe features of preeclampsia were diagnosed by the presence of one or more of the following findings: (1) systolic blood pressure > 160 mmHg or a diastolic blood pressure > 110 mmHg, (2) thrombocytopenia (platelet count less than 100,000/microliter), (3) impairment of liver function as indicated by abnormally elevated blood concentrations of liver enzymes (to twice normal concentration), severe persistent right upper quadrant or epigastric pain unresponsive to medication and not accounted for by alternative diagnoses, or both, (4) progressive renal insufficiency (serum creatinine concentration greater than 1.1 mg/dL or a doubling of the serum creatinine concentration in the absence of other renal disease), (5) pulmonary oedema, (6) new onset cerebral or visual disturbances [11,102,103,104].

The clinical presentation of hemolysis, elevated liver enzymes, and low platelet count (HELLP) syndrome is one of the more severe forms of preeclampsia. The following criteria to make the diagnosis are currently used: lactate dehydrogenase (LDH) elevated to 600 IU/L or more, aspartate aminotransferase (AST) and alanine aminotransferase (ALT) elevated more than twice the upper limit of normal, and the platelets count less than 100,000 × 10^9^/L. In HELLP syndrome, the main presenting symptoms are right upper quadrant pain, generalized malaise, nausea and vomiting [11,102,103,104].

Fetal growth restriction was diagnosed when the estimated fetal weight (EFW), calculated using the Hadlock formula (Astraia Software, version 1.25.2, GmbH, München, Deutschland), was below the tenth percentile for the evaluated gestational age, adjustments were made for the appropriate population standards of the Czech Republic. In addition to fetal weight below the threshold of the 10^th^ percentile FGR foetuses had at least one of the following pathological finding: an abnormal pulsatility index in the umbilical artery, absent or reversed end-diastolic velocity waveforms in the umbilical artery, an abnormal pulsatility index in the middle cerebral artery, a sign of a blood flow centralisation, and a deficiency of amniotic fluid (anhydramnios and oligohydramnios).

Centralization of the fetal circulation represents a protective reaction of the foetus against hypoxia that manifests itself in redistribution of the circulation in the brain, liver and heart at the expense of the flow reduction in the periphery [105,106]. The cerebroplacental ratio (CPR) quantifies redistribution of cardiac output by dividing Doppler indices from representative cerebral and fetoplacental vessels.

Patients with a complicated gestation demonstrating premature rupture of membranes, in utero infections, fetal anomalies or chromosomal abnormalities, and fetal demise in utero or stillbirth were excluded from the study.

Written informed consent was provided for all participants enrolled in the study. The study was approved by the Ethics Committee of the Institute for the Care of the Mother and Child, Prague, Czech Republic (grant no. AZV 16-27761A, Long-term monitoring of complex cardiovascular profile in the mother, foetus and offspring descending from pregnancy-related complications, date of approval: 28.5.2015) and by the Ethics Committee of the Third Faculty of Medicine, Prague, Czech Republic (grant no. AZV 16-27761A, Long-term monitoring of complex cardiovascular profile in the mother, foetus and offspring descending from pregnancy-related complications, date of approval: 27.3.2014).

The clinical characteristics of women with a history of normal and complicated pregnancies are presented in Table 8.

### 4.2. Assessment of Vascular Endothelial Function

Vascular endothelial function was assessed using the non-invasive PAT™ probe and the Endo PAT™2000 device (Itamar Medical Ltd., Caesarea, Israel), a new type of finger plethysmograph that measures the magnitude and dynamics of peripheral arterial tone (manufacturer´s instructions).

The patient was kept in comfortable and fully relaxed position, asked to refrain from talking throughout the entire study.

If possible, the index fingers were inserted into the probes. The blue foam anchor ring was placed on the adjacent finger to the one with the probe on, as far back as possible not to come in contact with the PAT probe. Both patients’ forearms were afterwards supported on the arm supports. Digital flow mediated dilation was assessed during reactive hyperemia using measurements from both arms–occluded side and control side.

Reactive Hyperemia Index (RHI), the post-to-pre occlusion signal ratio in the occluded side, normalized to the control side and further corrected for baseline vascular tone, was reported. The normal ratio value was above 1.67, while ≤ 1.67 was considered endothelial dysfunction (manufacturer´s instructions).

### 4.3. Blood Pressure Measurements

For detailed information please see our previous publication [26]. Standardized blood pressure measurements were performed following New AHA Recommendations for Blood Pressure Measurement [107]. Blood pressure was measured 3 times in the right arm after a 5-min rest period during which the participant sits using an automated device (OMRON M6W, Omron Healthcare Co., Kyoto, Japan). The average of the last 2 systolic and diastolic pressures was used for the data analyses.

### 4.4. BMI and Waist Circumference Measurements

For detailed information please see our previous publication [26]. Body weight was measured to the nearest 0.05–0.1 kg using an electronic scale and height was measured to the nearest 0.1 cm using a built-in stadiometer (calibrated balance scales, RADWAG WPT 100/200 OW, RADWAG, Prague, Czech Republic). BMI was calculated as weight divided by height squared. Waist circumference was measured to the nearest 0.1 cm using a measuring tape. Each measurement was taking twice. A waist measurement of ≥80 cm was considered as an indicator of the level of internal fat deposits.

### 4.5. Biological Sampling

For detailed information please see our previous publication [26]. Fasting blood samples were collected at the time of the study visit, 3 to 11 years postpartum, when comprehensive examination of patient was performed (blood pressure measurements, BMI and waist circumference measurements). Patients were told not to consume anything but water for 12 h leading up to the test (collection of whole peripheral blood samples was performed within 9.00 a.m. to 10.00 a.m. GMT). Total serum cholesterol, high-density lipoprotein (HDL) cholesterol, low-density lipoprotein (LDL) cholesterol, triglycerides, lipoprotein A Lp(a), high-sensitivity C-reactive protein (CRP), uric acid, and plasma homocysteine were analysed using standard laboratory methods at the Institute for the Care of the Mother and Child.

### 4.6. Gene Expression of Cardiovascular/Cerebrovascular diseAse Associated microRNAs in Whole Peripheral Blood

For detailed information please see our previous publications [28,88,89,90,95]. Epigenetic profiling of microRNAs known to be involved in the onset of diverse cardiovascular/cerebrovascular diseases (miR-1-3p, miR-16-5p, miR-17-5p, miR-20a-5p, miR-20b-5p, miR-21-5p, miR-23a-3p, miR-24-3p, miR-26a-5p, miR-29a-3p, miR-92a-3p, miR-100-5p, miR-103a-3p, miR-125b-5p, miR-126-3p, miR-130b-3p, miR-133a-3p, miR-143-3p, miR-145-5p, miR-146a-5p, miR-155-5p, miR-181a-5p, miR-195-5p, miR-199a-5p, miR-210-3p, miR-221-3p, miR-342-3p, miR-499a-5p, and miR-574-3p) was the subject of our interest. In brief, homogenized cell lysates were prepared immediately after collection of whole peripheral blood samples (EDTA tubes, 200 µl) using QIAamp RNA Blood Mini Kit (Qiagen, Hilden, Germany, no: 52304). Total RNA was extracted from homogenized cell lysates stored at −80 °C using a mirVana microRNA Isolation kit (Ambion, Austin, TX, USA, no: AM1560) and followed by an enrichment procedure for small RNAs. To minimize DNA contamination, the eluted RNA was treated for 30 min at 37 °C with 5 µL of DNase I (Thermo Fisher Scientific, Carlsbad, CA, USA, no: EN0521).

Each microRNA was reverse transcribed into cDNA by a TaqMan MicroRNA Assay (using the first generation chemistry involving miRNA-specific stem loop primer) and TaqMan MicroRNA Reverse Transcription Kit (Applied Biosystems, Branchburg, NJ, USA, no: 4366597) in a total reaction volume of 10 µL.

Further, 3 µL of cDNA were mixed with specific primers and the TaqMan MGB probes (TaqMan MicroRNA Assays, Applied Biosystems), and the ingredients of the TaqMan Universal PCR Master Mix (Applied Biosystems, no: 4318157) in a total reaction volume of 15 µL. TaqMan PCR conditions were set on 7500 Real-Time PCR System (Applied Biosystems) as described in the TaqMan guidelines.

The expression of microRNAs was determined using the comparative Ct method [108] relative to normalization factor [geometric mean of two selected endogenous controls - two non-coding small nucleolar RNAs (RNU58A and RNU38B)] [109]. A reference sample, RNA fraction highly enriched for small RNAs isolated from the fetal part of one randomly selected placenta derived from gestation with normal course, was used throughout the study for relative quantification.

### 4.7. Statistical Analysis

Data normality was assessed using the Shapiro-Wilk test [110]. The Chi-square test and univariate logistic regression model were used to compare the presence of vascular endothelial dysfunction across the individual groups (MedCalc Software bvba, Ostend, Belgium). Data are expressed as number (percent) of patients with the presence of vascular endothelial dysfunction. The significance level was established at *p*-value of *p* < 0.05.

ANOVA (Analysis of variance) and ANCOVA (Analysis of covariance) were used to test possible differences in mean concentrations of dependant variables (RHI) between particular groups of patients. We made adjustment for the age of patients (MedCalc Software bvba). Results were expressed as mean ± SE (standard error). The significance level was established at *p*-value of *p* < 0.05.

In case of ANOVA and ANCOVA significant results, Receivers operating characteristic (ROC) curves were constructed to calculate the area under the curve (AUC) and the best cut-off point was used in order to calculate the respective sensitivity at 90.0% specificity (MedCalc Software bvba). For every possible threshold or cut-off value, the MedCalc program reports the sensitivity, specificity, likelihood ratio positive (LR + ), likelihood ratio negative (LR-).

Correlation between variables was calculated using the Pearson correlation coefficient (r) and the Spearman’s rank correlation coefficient (*ρ*). The Pearson correlation coefficient was used for normally distributed variables and the Spearman’s rank correlation coefficient was used for variables with skewed distribution.

Mann-Whitney U test was used to compare microRNA expression between women with normal and abnormal vascular endothelial function. Data are expressed as median (25^th^percentile –75^th^ percentile). The significance level was established at a *p*-value of *p* < 0.05.

## 5. Conclusions

Causes of vascular endothelial dysfunction still need to be resolved in young and middle-aged population. Nevertheless, it is apparent that at least the lower RHI values occur more frequently in young and middle-aged women with higher BMI and central obesity, which are both considered as risk factors of later development of cardiovascular diseases.

Although no significant impact of the course of previous pregnancies on vascular endothelial function was found, consecutive large scale studies are needed to discover if there is any association between the occurrence of fetal growth restriction during gestation and vascular endothelial dysfunction in the mother.

Interestingly, epigenetic changes are induced in whole peripheral blood of young and middle-aged women by vascular endothelial dysfunction. A proportion of women with vascular endothelial dysfunction had aberrant expression profiles of microRNAs associated with cardiovascular and cerebrovascular diseases (miR-1-3p, miR-23a-3p, and miR-499a-5p). Lower RHI values also correlated with increased expression of miR-1-3p, miR-23a-3p, and miR-499a-5p in whole peripheral blood.

We suppose that screening of these particular microRNAs associated with vascular endothelial dysfunction may help to stratify a highly risky group of young and middle-aged women that would benefit from early implementation of primary prevention strategies.

Nevertheless, overall, it is obvious, that vascular endothelial dysfunction is one out of multiple cardiovascular risk factors which has only a partial impact on abnormal expression of cardiovascular and cerebrovascular disease associated microRNAs in whole peripheral blood of young and middle-aged women. That´s the reason, why we have been continuously working on the identification of all cardiovascular risk factors causing abnormal expression of cardiovascular and cerebrovascular disease associated microRNAs in whole peripheral blood of young and middle-aged women.

## 6. Patent

National Patent awarded—Industrial Property Office, Czech Republic (PV 2018-597).

## Figures and Tables

**Figure 1 ijms-21-00430-f001:**
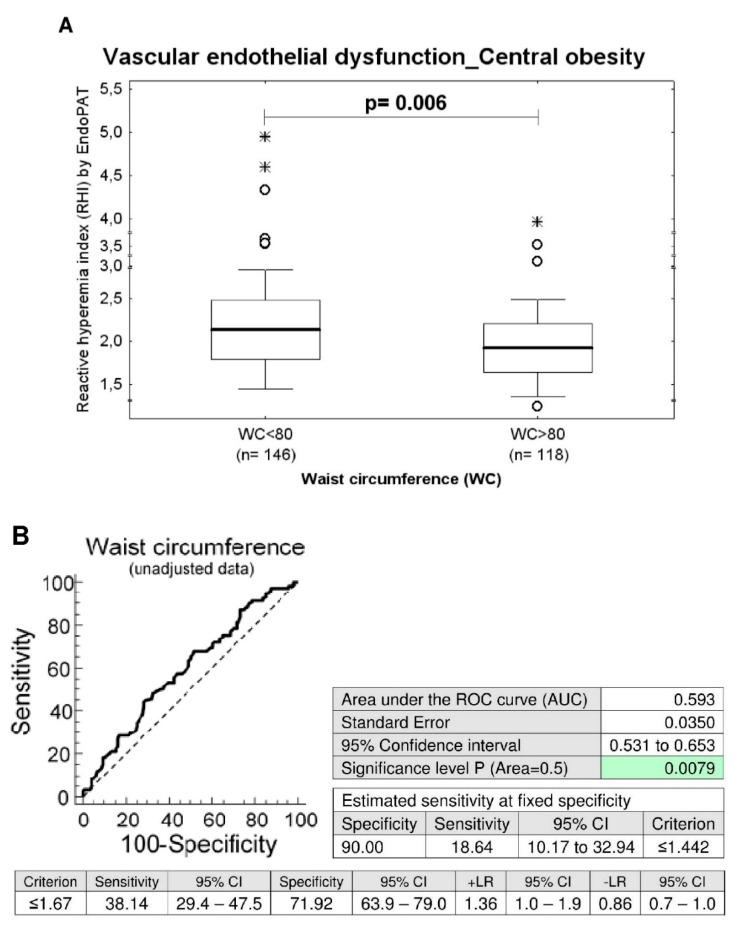
Association between the occurrence of vascular endothelial dysfunction and the central obesity. (**A**) Young and middle-aged women with central obesity (waist circumference values ≥ 80 cm) showed significantly decreased RHI levels when compared with the group of women with normal values of waist circumference (waist circumference values < 80 cm), (unadjusted data: *p* = 0.006, data adjusted for the age of patients: *p* = 0.008). (**B**) At 10.0% FPR, a markedly higher proportion of women with waist circumference values above 80 cm had RHI values below cut-off values. 18.64% women before adjustment of the data for the age had RHI values ≤ 1.442. In parallel, 21.19% women after adjustment of the data for the age had significantly lower RHI values.

**Figure 2 ijms-21-00430-f002:**
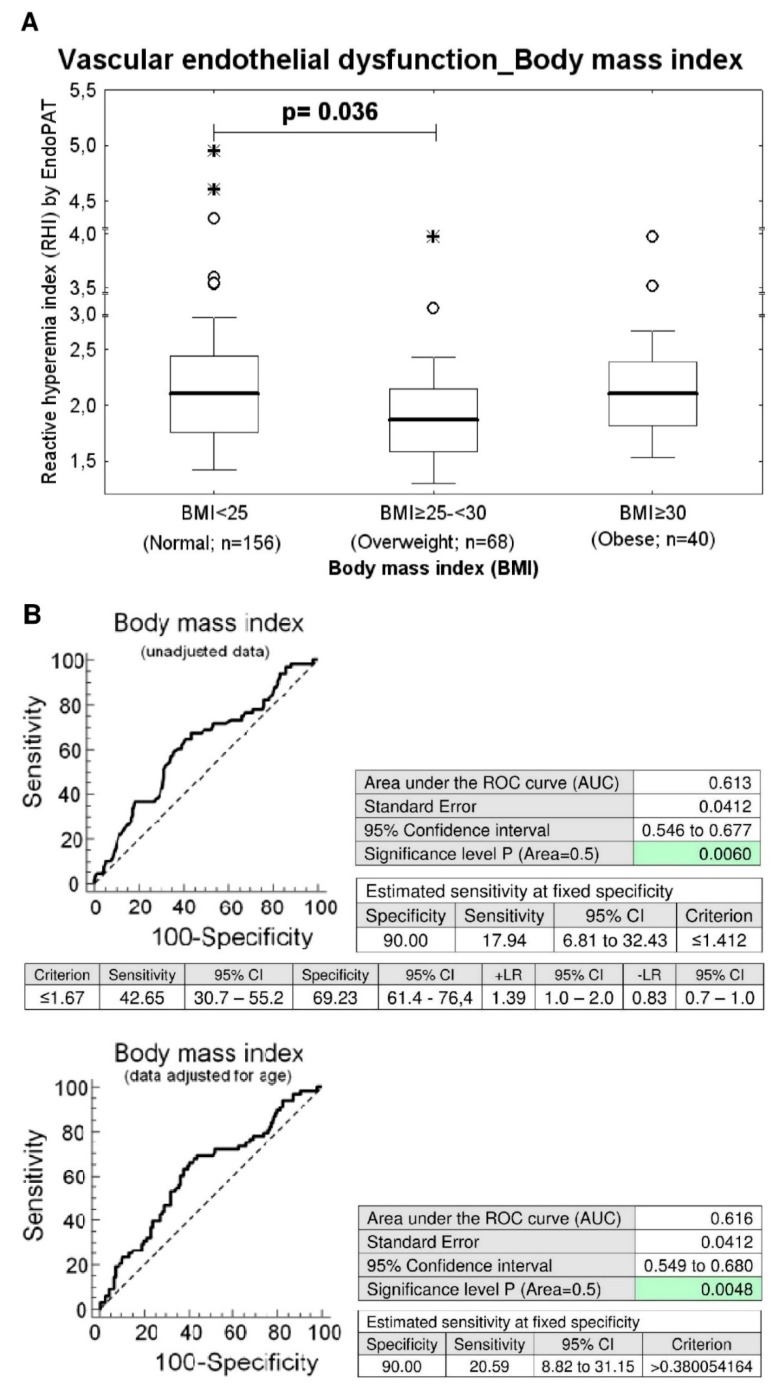
Association between the occurrence of vascular endothelial dysfunction and overweight. (**A**) Overweight women *(BMI* ≥ 25–<30) showed significantly decreased RHI levels when compared with the group of women with normal values of BMI (*BMI* < 25) (unadjusted data: *p* = 0.036, data adjusted for the age of women: *p* = 0.039). (**B**) At 10.0% FPR, a markedly higher proportion of overweight women had RHI values below cut-off values. 17.94% young women before adjustment of the data for the age had RHI values ≤ 1.412. In parallel, 20.59% women after adjustment of the data for the age had significantly lower RHI values.

**Figure 3 ijms-21-00430-f003:**
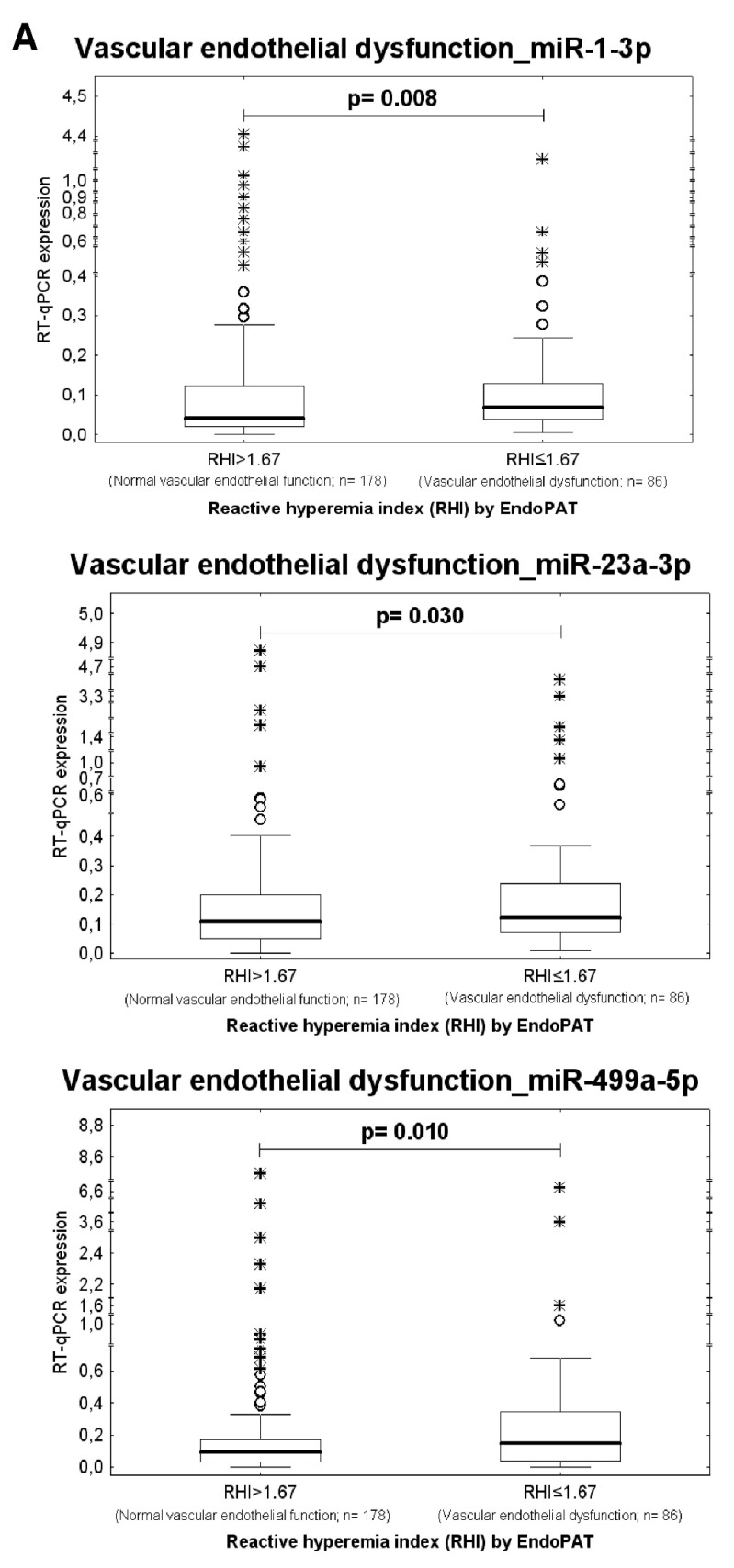
Higher expression rates of miR-1-3p, miR-23a-3p, miR-499a-5p in whole peripheral blood of young and middle-aged women with the presence of vascular endothelial dysfunction. (**A**) Higher expression rates of miR-1-3p, miR-23a-3p, and miR-499a-5p were detected in whole peripheral blood of young and middle-aged women with the occurrence of vascular endothelial dysfunction (RHI ≤ 1.67) when compared with those possessing normal vascular endothelial function (RHI > 1.67), respectively. (**B**) At 10.0% FPR, a proportion of women with the occurrence of vascular endothelial dysfunction was identified to have up-regulated expression profile of miR-1-3p (11.76%), miR-23a-3p (17.65%), and miR-499a-5p (18.82%).

**Figure 4 ijms-21-00430-f004:**
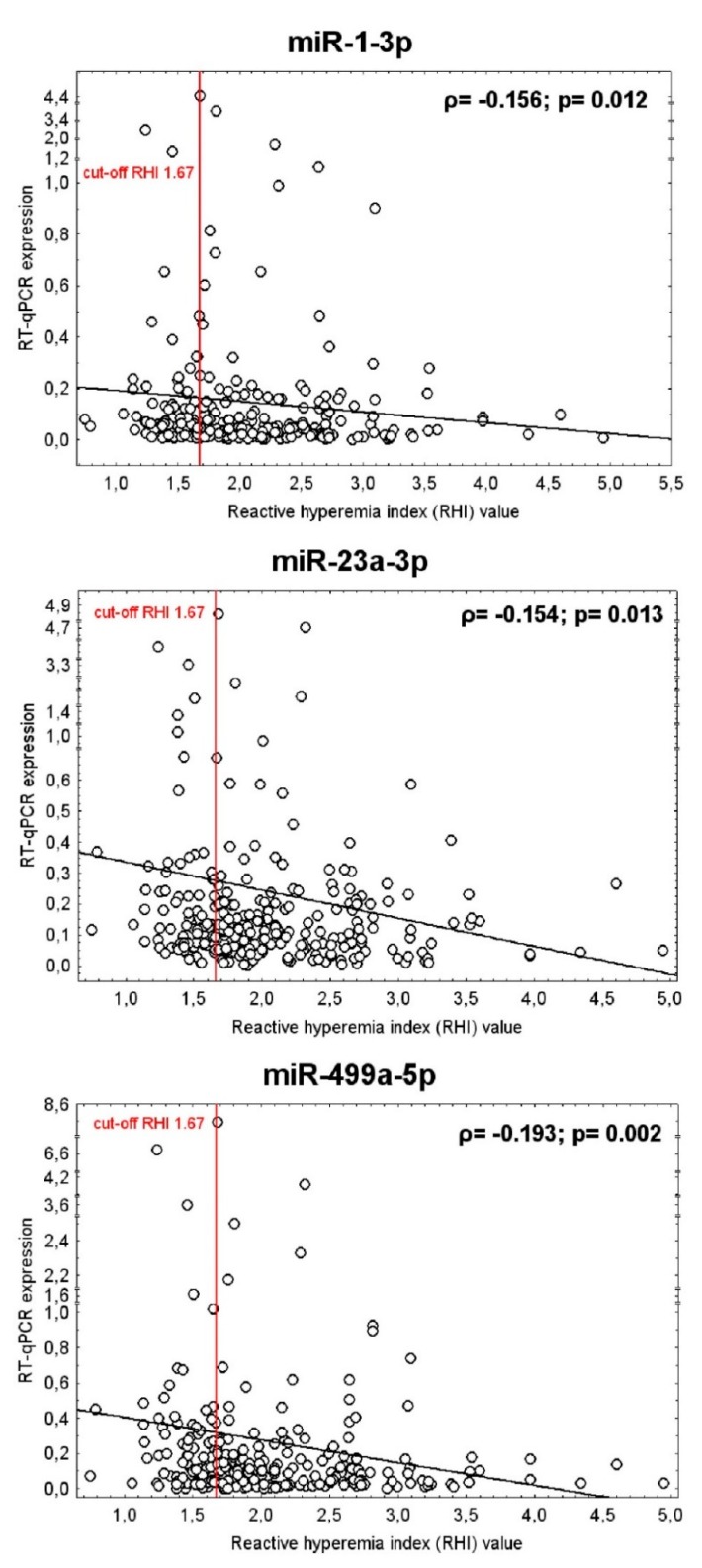
Association between the vascular endothelial function (RHI values) and the expression of miR-1-3p, miR-23a-3p, and miR-499a-5p in whole peripheral blood of young and middle-aged women. Correlation between variables was calculated using the Spearman’s rank correlation coefficient (*ρ*). Spearman’s rank correlation coefficient, a nonparametric measure of rank correlation, assesses how well the relationship between two variables can be described using a monotonic function. The significance level was established at a *p*-value of *p* < 0.05. The RHI values showed a week negative correlation with the expression of miR-1-3p, miR-23a-3p, and miR-499a-5p in whole peripheral blood of young and middle-aged women. That means that women with a finding of lower RHI values showed significantly increased levels of miR-1-3p, miR-23a-3p, and miR-499a-5p in whole peripheral blood, respectively.

**Table 1 ijms-21-00430-t001:** Impact of classical cardiovascular risk factors and a history of pregnancy on vascular endothelial function_ overview (ANOVA, ANCOVA).

	Endothelial Function (RHI) Mean (SE)		*p* Value
On blood pressure treatment [no (*n* = 253) vs. yes (*n* = 11)]	Unadjusted data	2.046 (0.041) vs. 1.916 (0.195)	*p* = 0.516
	Adjusted data	2.046 (0.041) vs. 1.921 (0.195)	*p* = 0.531
Trombophilic gene mutations [no (*n* = 246) vs. yes (*n* = 18)]	Unadjusted data	2.042 (0.041) vs. 2.027 (0.153)	*p* = 0.924
	Adjusted data	2.041 (0.041) vs. 2.036 (0.153)	*p* = 0.974
Current smoking of cigarettes [non-smokers + ex-smokers (*n* = 228) vs. smokers (*n* = 36)]	Unadjusted data	2.028 (0.043) vs. 2.119 (0.108)	*p* = 0.435
	Adjusted data	2.029 (0.043) vs. 2.113 (0.108)	*p* = 0.476
BMI [normal (*n* = 156) vs. abnormal (*n* = 108)] (<25 vs. ≥25)	Unadjusted data	2.101 (0.052) vs. 1.953 (0.062)	*p* = 0.068
	Adjusted data	2.099 (0.052) vs. 1.957 (0.062)	*p* = 0.082
BMI [normal (*n* = 156) vs. overweight (*n* = 68) vs. obese (*n* = 40)] (<25 vs. ≥ 25–29.9 vs. ≥30)	Unadjusted data	2.101 (0.051) vs. 1.866 (0.078) vs. 2.102 (0.101)	Normal BMI vs. overweight *p* = 0.036 ↓ RHI in overweight women Normal BMI vs. obese *p* = 1.0
	Adjusted data	2.099 (0.051) vs. 1.865 (0.078) vs. 2.117 (0.102)	Normal BMI vs. overweight *p* = 0.039 ↓ RHI in overweight women Normal BMI vs. obese *p* = 1.0
Waist circumference [normal (*n* = 146) vs. abnormal (*n* = 118)] (<80 cm vs. ≥ 80 cm)	Unadjusted data	2.138 (0.053) vs. 1.920 (0.059)	*p* = 0.006 ↓ RHI in women with waist circumference ≥ 80 cm
	Adjusted data	2.135 (0.053) vs. 1.924 (0.059)	*p* = 0.008 ↓ RHI in women with waist circumference ≥ 80 cm
SBP [normal (*n* = 241) vs. abnormal (*n* = 23)] (<140 mmHg vs. ≥ 140 mmHg)	Unadjusted data	2.033 (0.042) vs. 2.123 (0.135)	*p* = 0.526
	Adjusted data	2.032 (0.042) vs. 2.134 (0.135)	*p* = 0.471
SBP [normal (*n* = 138) vs. prehypertension (*n* = 103) vs. hypertension (*n* = 23)] (<120 mmHg vs. ≥120–139 mmHg vs. ≥ 140 mmHg)	Unadjusted data	2.030 (0.055) vs. 2.037 (0.064) vs. 2.123 (0.135)	Normal SBP vs. prehypertension *p* = 1.0 Normal SBP vs. hypertension *p* = 1.0
	Adjusted data	2.029 (0.055) vs. 2.035 (0.064) vs. 2.134 (0.136)	Normal SBP vs. prehypertension *p* = 1.0 Normal SBP vs. hypertension *p* = 1.0
DBP [normal (*n* = 229) vs. abnormal (*n* = 35)] (< 90 mmHg vs. ≥ 90 mmHg)	Unadjusted data	2.035 (0.043) vs. 2.077 (0.110)	*p* = 0.721
	Adjusted data	2.034 (0.043) vs. 2.083 (0.110)	*p* = 0.678
DBP [normal (*n* = 172) vs. prehypertension (*n* = 57) vs. hypertension (*n* = 35)] (< 80 mmHg vs. ≥80–89 mmHg vs. ≥ 90 mmHg)	Unadjusted data	2.019 (0.050) vs. 2.084 (0.086) vs. 2.077 (0.110)	Normal DBP vs. prehypertension *p* = 1.0 Normal DBP vs. hypertension *p* = 1.0
	Adjusted data	2.020 (0.050) vs. 2.076 (0.086) vs. 2.083 (0.110)	Normal DBP vs. prehypertension *p* = 1.0 Normal DBP vs. hypertension *p* = 1.0
Serum total cholesterol [normal (*n* = 125) vs. abnormal (*n* = 139)] (≤ 5 mmol/L vs. > 5 mmol/L)	Unadjusted data	2.004 (0.058) vs. 2.074 (0.055)	*p* = 0.386
	Adjusted data	1.999 (0.058) vs. 2.078 (0.055)	*p* = 0.329
Serum HDL cholesterol [normal (*n* = 230) vs. abnormal (*n* = 34)] (≥ 1.2 mmol/L vs. <1.2 mmol/L)	Unadjusted data	2.046 (0.043) vs. 2.002 (0.111)	*p* = 0.710
	Adjusted data	2.046 (0.043) vs. 2.006 (0.111)	*p* = 0.741
Serum LDL cholesterol [normal (*n* = 100) vs. abnormal (*n* = 164)] (≤ 3 mmol/L vs. > 3 mmol/L)	Unadjusted data	1.983 (0.065) vs. 2.074 (0.051)	*p* = 0.267
	Adjusted data	1.976 (0.065) vs. 2.079 (0.051)	*p* = 0.217
Serum triglycerides [normal (*n* = 248) vs. abnormal (*n* = 16)] (≤ 1.7 mmol/L vs. > 1.7 mmol/L)	Unadjusted data	2.039 (0.041) vs. 2.075 (0.162)	*p* = 0.827
	Adjusted data	2.038 (0.041) vs. 2.083 (0.162)	*p* = 0.786
Serum Lp(a) [normal (*n* = 212) vs. abnormal (*n* = 52)] (≤ 72.0 nmol/L vs. > 72.0 nmol/L)	Unadjusted data	2.032 (0.045) vs. 2.076 (0.090)	*p* = 0.665
	Adjusted data	2.031 (0.045) vs. 2.082 (0.090)	*p* = 0.609
Serum CRP [normal (*n* = 223) vs. abnormal (*n* = 41)] (≤ 5 mg/L vs. > 5 mg/L)	Unadjusted data	2.032 (0.043) vs. 2.089 (0.101)	*p* = 0.608
	Adjusted data	2.032 (0.043) vs. 2.088 (0.101)	*p* = 0.613
Plasma homocysteine [normal (*n* = 229) vs. abnormal (*n* = 35)] (≤ 13.6 μmol/L vs. > 13.6 μmol/L)	Unadjusted data	2.054 (0.043) vs. 1.954 (0.109)	*p* = 0.397
	Adjusted data	2.054 (0.043) vs. 1.954 (0.109)	*p* = 0.394
Serum uric acid [normal (*n* = 233) vs. abnormal (*n* = 31)] (≤339 μmol/L vs. > 339 μmol/L)	Unadjusted data	2.064 (0.042) vs. 1.870 (0.116)	*p* = 0.117
	Adjusted data	2.063 (0.042) vs. 1.877 (0.116)	*p* = 0.134
Current hormonal contraceptive use [non-users + ex-users (*n* = 194) vs. users (*n* = 70)]	Unadjusted data	2.038 (0.047) vs. 2.049 (0.077)	*p* = 0.901
	Adjusted data	2.041 (0.047) vs. 2.041 (0.078)	*p* = 0.991
Total number of pregnancies per patient [1 (*n* = 61) vs. 2 (*n* = 115) vs. 3+ (*n* = 88)]	Unadjusted data	2.050 (0.083) vs. 2.117 (0.061) vs. 1.935 (0.069)	1 pregnancy vs. 2 pregnancies *p* = 1.0 1 pregnancy vs. 3+ pregnancies *p* = 0.857
	Adjusted data	2.044 (0.083) 2.114 (0.061) vs. 1.943 (0.071)	1 pregnancy vs. 2 pregnancies *p* = 1.0 1 pregnancy vs. 3+ pregnancies *p* = 1.0
Total parity per patient [1 (*n* = 77) vs. 2 (*n* = 150) vs. 3+ (*n* = 37)]	Unadjusted data	2.038 (0.074) vs. 2.071 (0.053) vs. 1.927 (0.107)	1 child vs. 2 children *p* = 1.0 1 child vs. 3+ children *p* = 1.0
	Adjusted data	2.038 (0.074) vs. 2.066 (0.053) vs. 1.943 (0.109)	1 child vs. 2 children *p* = 1.0 1 child vs. 3+ children *p* = 1.0
Infertility treatment [no (*n* = 226) vs. yes (*n* = 38)]	Unadjusted data	2.021 (0.043) vs. 2.161 (0.105)	*p* = 0.216
	Adjusted data	2.017 (0.043) vs. 2.183 (0.106)	*p* = 0.149

Data are expressed as mean (SE; standard error). Analysis of variance (ANOVA) was used for unadjusted data and Analysis of covariance (ANCOVA) for adjusted data. ANCOVA data were adjusted for the age of women. The significance level was established at *p*-value of *p* < 0.05 (Bonferroni corrected *p*-values). Statistically significant results are marked in bold. Women with central obesity (waist circumference values ≥ 80 cm) showed significantly decreased RHI levels when compared with the group of women with normal values of waist circumference (waist circumference values < 80 cm). Overweight women (*BMI* ≥25 –<30) showed significantly decreased RHI levels when compared with the group of women with normal values of BMI (*BMI* < 25). BMI, body mass index; CRP, C-reactive protein; DBP, diastolic blood pressure; HDL, high-density lipoprotein; LDL, low-density lipoprotein; Lp(a), lipoprotein a; SBP, systolic blood pressure; RHI, Reactive Hyperemia Index.

**Table 2 ijms-21-00430-t002:** Impact of classical cardiovascular risk factors and a history of pregnancy on the prevalence of vascular endothelial dysfunction_ overview (logistic regression analyses).

Prevalence of Vascular Endothelial Dysfunction	Group 1 (%)	Group 2 (%)	*p* Value	OR (95% CI)
On blood pressure treatment [no (*n* = 253) vs. yes (*n* = 11)]	83 (32.81%)	3 (27.27%)	0.702	0.768 (0.199–2.971)
Trombophilic gene mutations [no (*n* = 246) vs. yes (*n* = 18)]	80 (32.52%)	6 (33.33%)	0.943	1.038 (0.376–2.865)
Current smoking of cigarettes [non-smokers + ex-smokers (*n* = 228) vs. smokers (*n* = 36)]	78 (34.21%)	8 (22.22%)	0.158	0.550 (0.239–1.263)
BMI [normal (*n* = 156) vs. abnormal (*n* = 108)] (<25 vs. ≥25)	48 (30.77%)	38 (35.19%)	0.452	1.221 (0.725–2.057)
BMI				
[normal (*n* = 156) vs. overweight (*n* = 68)] (<25 vs. ≥25–29.9)	48 (30.77%)	29 (42.65%)	0.087	1.673 (0.929–3.014)
[normal (*n* = 156) vs. obese (*n* = 40)] (<25 vs. ≥30)	48 (30.77%)	9 (22.50%)	0.307	0.653 (0.289–1.478)
Waist circumference [normal (*n* = 146) vs. abnormal (*n* = 118)] (<80 cm vs. ≥ 80 cm)	41 (28.08%)	45 (38.14%)	0.084	1.579 (0.941–2.650)
SBP [normal (*n* = 241) vs. abnormal (*n* = 23)] (<140 mmHg vs. ≥140 mmHg)	82 (34.02%)	4 (17.39%)	0.114	0.408 (0.134–1.240)
SBP				
[normal (*n* = 138) vs. prehypertension (*n* = 103)] (<120 mmHg vs. ≥120–139 mmHg)	49 (35.51%)	33 (32.04%)	0.574	0.856 (0.498–1.471)
[normal (*n* = 138) vs. hypertension (*n* = 23)] (<120 mmHg vs. ≥140 mmHg)	49 (35.51%)	4 (17.39%)	0.096	0.382 (0.123–1.188)
DBP [normal (*n* = 229) vs. abnormal (*n* = 35)] (<90 mmHg vs. ≥90 mmHg)	79 (34.50%)	7 (20.0%)	0.094	0.475 (0.199–1.135)
DBP				
[normal (*n* = 172) vs. prehypertension (*n* = 57)] (<80 mmHg vs. ≥80–89 mmHg)	62 (36.05%)	17 (29.82%)	0.393	0.754 (0.395–1.440)
[normal (*n* = 172) vs. hypertension (*n* = 35)] (<80 mmHg vs. ≥90 mmHg)	62 (36.05%)	7 (20.00%)	0.072	0.444 (0.183–1.075)
Serum total cholesterol [normal (*n* = 125) vs. abnormal (*n* = 139)] (≤5 mmol/L vs. >5 mmol/L)	39 (31.30%)	47 (33.81%)	0.651	1.127 (0.672–1.888)
Serum HDL cholesterol [normal (*n* = 230) vs. abnormal (*n* = 34)] (≥1.2 mmol/L vs. <1.2 mmol/L)	75 (32.61%)	11 (32.35)	0.976	0.988 (0.458–2.134)
Serum LDL cholesterol [normal (*n* = 100) vs. abnormal (*n* = 164)] (≤3 mmol/L vs. >3 mmol/L)	32 (32.0%)	54 (32.91%)	0.808	1.068 (0.628–1.817)
Serum triglycerides [normal (*n* = 248) vs. abnormal (*n* = 16)] (≤1.7 mmol/L vs. >1.7 mmol/L)	81 (32.66%)	5 (31.25%)	0.907	0.937 (0.315–2.787)
Serum Lp(a) [normal (*n* = 212) vs. abnormal (*n* = 52)] (≤72.0 nmol/L vs. >72.0 nmol/L)	71 (33.49%)	15 (28.85%)	0.522	0.805 (0.414–1.564)
Serum CRP [normal (*n* = 223) vs. abnormal (*n* = 41)] (≤5 mg/L vs. >5 mg/L)	75 (33.63%)	11 (26.83%)	0.974	0.991 (0.586–1.678)
Plasma homocysteine [normal (*n* = 229) vs. abnormal (*n* = 35)] (≤13.6 μmol/L vs. >13.6 μmol/L)	72 (31.44%)	14 (40.0%)	0.316	1.454 (0.700–3.021)
Serum uric acid [normal (*n* = 233) vs. abnormal (*n* = 31)] (≤339 μmol/L vs. >339 μmol/L)	72 (30.90%)	14 (45.16%)	0.115	1.842 (0.861–3.938)
Current hormonal contraceptive use [non-users + ex-users (*n* = 194) vs. users (*n* = 70)]	64 (32.99%)	22 (31.43%)	0.811	0.931 (0.518–1.674)
Total number of pregnancies per patient				
[1 (*n* = 61) vs. 2 (*n* = 115)]	17 (27.87%)	32 (27.83%)	0.995	0.998 (0.499–1.995)
[1 (*n* = 61) vs. 3+ (*n* = 88)]	17 (27.87%)	37 (42.05%)	0.078	1.878 (0.931–3.788)
Total parity per patient				
[1 (*n* = 77) vs. 2 (*n* = 150)]	24 (31.17%)	49 (32.67%)	0.819	1.071 (0.593–1.934)
[1 (*n* = 77) vs. 3+ (*n* = 37)]	24 (31.17%)	13 (35.14%)	0.672	1.196 (0.522–2.742)
Infertility treatment [no (*n* = 226) vs. yes (*n* = 38)]	75 (33.19%)	11 (28.95%)	0.606	0.820 (0.386–1.743)

Data are expressed as number (percent) of patients with the presence of vascular endothelial dysfunction. Logistic regression was used to compare the presence of vascular endothelial dysfunction between particular groups. The significance level was established at *p*-value of *p* < 0.05. No impact of any factor on the occurrence of vascular endothelial dysfunction was found. BMI, body mass index; CI, confidence intervals; CRP, C-reactive protein; DBP, diastolic blood pressure; HDL, high-density lipoprotein; LDL, low-density lipoprotein; Lp(a), lipoprotein a; RHI, Reactive Hyperemia Index; SBP, systolic blood pressure; OR, odds ratios.

**Table 3 ijms-21-00430-t003:** Association between the vascular endothelial function (RHI values) and classical cardiovascular risk factors and a history of pregnancy (Pearson correlation coefficient, Spearman’s coefficient of rank correlation).

*RHI (Non-Normal Distribution)*	Data Distribution	Pearson Correlation Coefficient, *p* Value	Spearman’s Coefficient of Rank Correlation (rho), *p* Value
age	*Non-normal distribution*	-	*ρ* = −0.046, *p* = 0.457
BMI	*Non-normal distribution*	-	*ρ* = −0.063, *p* = 0.307
Waist circumference	*Non-normal distribution*	-	*ρ* = −0.081, *p* = 0.188
SBP	*Non-normal distribution*	-	*ρ* = 0.079, *p* = 0.199
DBP	*Non-normal distribution*	-	*ρ* = 0.044, *p* = 0.478
Heart rate at rest	*Non-normal distribution*	-	*ρ* = 0.051, *p* = 0.407
Serum total cholesterol	*Non-normal distribution*	-	*ρ* = 0.015, *p* = 0.810
Serum HDL cholesterol	*Non-normal distribution*	-	*ρ* = −0.030, *p* = 0.631
Serum LDL cholesterol	*Non-normal distribution*	-	*ρ* = −0.002, *p* = 0.973
Serum triglycerides	*Non-normal distribution*	-	*ρ* = −0.008, *p* = 0.903
Serum Lp(a)	*Non-normal distribution*	-	*ρ* = −0.002, *p* = 0.981
Serum CRP	*Non-normal distribution*	-	*ρ* = 0.057, *p* = 0.357
Plasma homocysteine	*Non-normal distribution*	-	*ρ* = −0.020, *p* = 0.753
Serum uric acid	*Normal distribution*	*r* = −0.111, *p* = 0.072	*ρ* = −0.111, *p* = 0.072
Time elapsed since the delivery	*Non-normal distribution*	-	*ρ* = 0.014, *p* = 0.816
Total number of pregnancies per patient	*Non-normal distribution*	-	*ρ* = −0.114, *p* = 0.064
Total parity per patient	*Non-normal distribution*	-	*ρ* = −0.037, *p* = 0.548

Correlation between variables was calculated using the Pearson correlation coefficient (r) and the Spearman’s rank correlation coefficient (*ρ*). The Pearson correlation coefficient was used for normally distributed variables and the Spearman’s rank correlation coefficient was used for variables with skewed distribution. The significance level was established at a *p*-value of *p* < 0.05. The RHI values did not correlate with any factor. BMI, body mass index; CRP, C-reactive protein; DBP, diastolic blood pressure; HDL, high-density lipoprotein; LDL, low-density lipoprotein; Lp(a), lipoprotein a; SBP, systolic blood pressure; RHI, Reactive Hyperemia Index.

**Table 4 ijms-21-00430-t004:** Impact of a history of pregnancy-related complications (GH, PE, FGR) with respect to the severity of the disease and delivery date on vascular endothelial function (RHI values)_ overview (ANOVA and ANCOVA analyses)

		**NTP** **(*n* = 74)**	**GH** **(*n* = 48)**	**PE** **(*n* = 114)**	**FGR** **(*n* = 28)**	**Diagnostic *Groups*** **(Normal vs. Diseased)**	***p* Value** **(ANOVA, ANCOVA)**
RHI	Unadjusted data	2.066 (0.075)	1.907 (0.093)	2.094 (0.060)	1.989 (0.122)	NTP vs. GH	*p* = 1.0
						NTP vs. PE	*p* = 1.0
						NTP vs. FGR	*p* = 1.0
	Adjusted data	2.068 (0.075) ^A^	1.910 (0.093) ^A^	2.091 (0.060) ^A^	1.987 (0.122) ^A^	NTP vs. GH	*p* = 1.0
						NTP vs. PE	*p* = 1.0
						NTP vs. FGR	*p* = 1.0
		**NTP** **(*n* = 74)**		**PE w/o SF** **(*n* = 28)**	**PE w SF** **(*n* = 86)**	**Diagnostic** ***Groups*** **(Normal vs. Diseased)**	***p* Value** **(ANOVA, ANCOVA)**
RHI	Unadjusted data	2.066 (0.075)		2.200 (0.121)	2.059 (0.069)	NTP vs. PE w/o SF	*p* = 1.0
						NTP vs. PE w SF	*p* = 1.0
	Adjusted data	2.066 (0.075) ^A^		2.201 (0.122) ^A^	2.059 (0.069) ^A^	NTP vs. PE w/o SF	*p* = 1.0
						NTP vs. PE w SF	*p* = 1.0
		**NTP** **(*n* = 74)**		**Early PE** **(*n* = 40)**	**Late PE** **(*n* = 74)**	**Diagnostic** ***Groups*** **(Normal vs. Diseased)**	***p* Value** **(ANOVA, ANCOVA)**
RHI	Unadjusted data	2.066 (0.075)		2.074 (0.102)	2.104 (0.075)	NTP vs. early PE	*p* = 1.0
						NTP vs. late PE	*p* = 1.0
	Adjusted data	2.066 (0.075) ^A^		2.074 (0.102) ^A^	2.105 (0.075) ^A^	NTP vs. early PE	*p* = 1.0
						NTP vs. late PE	*p* = 1.0
		**NTP** **(*n* = 74)**		**PE w/o HELLP** **(*n* = 98)**	**PE w HELLP** **(*n* = 16)**	**Diagnostic** ***Groups*** **(Normal vs. Diseased)**	***p* Value** **(ANOVA, ANCOVA)**
RHI	Unadjusted data	2.066 (0.075)		2.067 (0.065)	2.256 (0.160)	NTP vs. PE w/o HELLP	*p* = 1.0
						NTP vs. PE w HELLP	*p* = 0.848
	Adjusted data	2.066 (0.075) ^A^		2.067 (0.065) ^A^	2.256 (0.161) ^A^	NTP vs. PE w/o HELLP	*p* = 1.0
						NTP vs. PE w HELLP	*p* = 0.852
		**NTP** **(*n* = 74)**		**PE w/o FGR** **(*n* = 98)**	**PE w FGR** **(*n* = 16)**	**Diagnostic** ***Groups*** **(Normal vs. Diseased)**	***p* Value** **(ANOVA, ANCOVA)**
RHI	Unadjusted data	2.066 (0.075)		2.104 (0.065)	2.029 (0.161)	NTP vs. PE w/o FGR	*p* = 1.0
						NTP vs. PE w FGR	*p* = 1.0
	Adjusted data	2.066 (0.075) ^A^		2.104 (0.065) ^A^	2.028 (0.162) ^A^	NTP vs. PE w/o FGR	*p* = 1.0
						NTP vs. PE w FGR	*p* = 1.0
		**NTP** **(*n* = 74)**		**Early FGR** **(*n* = 9)**	**Late FGR** **(*n* = 19)**	**Diagnostic** ***Groups*** **(Normal vs. Diseased)**	***p* Value** **(ANOVA, ANCOVA)**
RHI	Unadjusted data	2.066 (0.085)		2.038 (0.243)	1.965 (0.168)	NTP vs. early FGR	*p* = 1.0
						NTP vs. late FGR	*p* = 1.0
	Adjusted data	2.065 (0.085) ^A^		2.037 (0.245) ^A^	1.969 (0.169) ^A^	NTP vs. early FGR	*p* = 1.0
						NTP vs. late FGR	*p* = 1.0

Data are expressed as mean (SE; standard error). Analysis of variance (ANOVA) was used for unadjusted data and Analysis of covariance (ANCOVA) for adjusted data. ^A^ ANCOVA data were adjusted for the age of women. The significance level was established at *p*-value of *p* < 0.05 (Bonferroni corrected *p*-values). No significant difference in RHI values was observed between the groups of women with a history of pregnancy-related complications (GH, PE, and FGR) and normally progressing pregnancies. No difference in RHI values was found between the control group and the groups of women with a history of PE w/o SF, PE w SF, early PE, late PE, PE w/o HELLP, PE w HELLP, PE w/o FGR, PE w FGR, early FGR, and late FGR. FGR, fetal growth restriction; GH, gestational hypertension; HELLP, hemolysis, elevated liver enzymes and low platelet count; PE, preeclampsia; PE w/o SF, preeclampsia without severe features; PE w/SF, preeclampsia with severe features; NTP, normotensive term pregnancies; RHI, Reactive Hyperemia Index.

**Table 5 ijms-21-00430-t005:** Impact of a history of pregnancy-related complications (GH, PE, FGR) with respect to the severity of the disease and delivery date on prevalence of vascular endothelial dysfunction_ overview (logistic regression analyses).

Prevalence of Vascular Endothelial Dysfunction	Case, *n* (%)	NTP, *n* (%)	*p* Value	OR (95% CI)
Pregnancy-related complications irrespective of the severity of the disease				
PE	29 (25.44%)	24 (32.43%)	0.299	0.711 (0.373–1.353)
GH	18 (37.50%)	24 (32.43%)	0.565	1.250 (0.584–2.674)
FGR	15 (53.57%)	24 (32.43%)	0.053	2.404 (0.989–5.842)
Women with a history of FGR with respect to the disease severity				
Early FGR	6 (66.67%)	24 (32.43%)	0.057	4.167 (0.959–18.102)
Late FGR	9 (47.37%)	24 (32.43%)	0.229	1.875 (0.674–5.219)
Women with a history of PE with respect to the disease severity				
PE w/o SF	6 (21.43%)	24 (32.43%)	0.280	0.568 (0.204–1.585)
PE w SF	23 (26.74%)	24 (32.43%)	0.432	0.761 (0.385–1.504)
Early PE	13 (32.50%)	24 (32.43%)	0.994	1.003 (0.441–2.281)
Late PE	16 (21.62%)	24 (32.43%)	0.141	0.575 (0.275–1.201)
PE w/o HELLP	27 (27.55%)	24 (32.43%)	0.488	0.792 (0.410–1.530)
PE w HELLP	2 (12.50%)	24 (32.43%)	0.128	0.298 (0.063–1.416)
PE w/o FGR	24 (24.49%)	24 (32.43%)	0.251	0.676 (0.346–1.320)
PE w FGR	5 (31.25%)	24 (32.43%)	0.927	0.947 (0.296–3.032)

Data are expressed as number (percent) of patients with the presence of endothelial dysfunction. Logistic regression was used to compare the presence of endothelial dysfunction between particular groups. The significance level was established at *p*-value of *p* < 0.05. No significant difference in the prevalence of endothelial dysfunction was observed between the groups of women with a history of pregnancy-related complications (GH, PE, and FGR) and normally progressing pregnancies. No difference in the prevalence of endothelial dysfunction between the control group and the groups of women with a history of PE w/o SF, PE w SF, early PE, late PE, PE w/o HELLP, PE w HELLP, PE w/o FGR, PE w FGR, early FGR, and late FGR was found. CI, confidence intervals; FGR, fetal growth restriction; GH, gestational hypertension; HELLP, hemolysis, elevated liver enzymes and low platelet count; NTP, normotensive term pregnancies; PE, preeclampsia PE w/o SF, preeclampsia without severe features; PE w/SF, preeclampsia with severe features; OR, odds ratios.

**Table 6 ijms-21-00430-t006:** Association between the occurrence of vascular endothelial dysfunction and the expression of cardiovascular and cerebrovascular disease associated microRNAs in whole peripheral blood of young and middle-aged women_ overview (Mann-Whitney U test analyses).

microRNA	Normal Endothelial Function RHI > 1.67 (*n* = 178) Median (25^th^ Percentile–75^th^ Percentile)	Endothelial Dysfunction RHI ≤ 1.67 (*n* = 86) Median (25^th^ Percentile–75^th^ Percentile)	*p* Value
miR-1-3p	4.170 (1.970–12.300) × 10^−2^	6.840 (3.780–13.000) × 10^−2^	*p* = 0.008
miR-16-5p	0.868 (0.625–1.249)	0.933 (0.708–1.147)	*p* = 0.548
miR-17-5p	1.044 (0.674–1.664)	1.174 (0.820–1.669)	*p* = 0.421
miR-20a-5p	1.085 (0.550–1.633)	0.995 (0.549–1.521)	*p* = 0.672
miR-20b-5p	1.051 (0.595–1.535)	1.111 (0.564–1.592)	*p* = 0.677
miR-21-5p	0.180 (0.101–0.293)	0.200 (0.133–0.295)	*p* = 0.215
miR-23a-3p	0.109 (0.051–0.200)	0.123 (0.073–0.239)	*p* = 0.030
miR-24-3p	0.204 (0.139–0.293)	0.213 (0.148–0.298)	*p* = 0.726
miR-26a-5p	0.346 (0.188–0.542)	0.406 (0.248–0.536)	*p* = 0.163
miR-29a-3p	0.187 (0.112–0.335)	0.211 (0.119–0.388)	*p* = 0.460
miR-92a-3p	1.759 (1.208–2.560)	1.704 (1.144–2.589)	*p* = 0.759
miR-100-5p	0.975 (0.554–1.880) × 10^−3^	1.000 (0.514–1.720) × 10^−3^	*p* = 0.950
miR-103a-3p	0.759 (0.397–1.294)	0.944 (0.493–1.365)	*p* = 0.095
miR-125b-5p	0.257 (0.121–0.447) × 10^−2^	0.254 (0.140–0.449) × 10^−2^	*p* = 0.808
miR-126-3p	0.178 (0.097–0.279)	0.191 (0.115–0.279)	*p* = 0.442
miR-130b-3p	0.345 (0.177–0.677)	0.379 (0.251–0.698)	*p* = 0.301
miR-133a-3p	7.440 (2.840–15.500) × 10^−2^	7.890 (3.870–18.600) × 10^−2^	*p* = 0.221
miR-143-3p	1.320 (0.614–2.920) × 10^−2^	1.430 (0.792–2.990) × 10^−2^	*p* = 0.485
miR-145-5p	7.450 (4.710–11.100) × 10^−2^	7.860 (4.630–12.000) × 10^−2^	*p* = 0.758
miR-146a-5p	0.844 (0.482–1.267)	0.845 (0.490–1.240)	*p* = 0.997
miR-155-5p	0.962 (0.680–1.448)	1.129 (0.731–1.599)	*p* = 0.353
miR-181a-5p	0.153 (0.086–0.245)	0.161 (0.093–0.254)	*p* = 0.708
miR-195-5p	3.240 (1.150–8.860) × 10^−2^	3.770 (1.360–9.730) × 10^−2^	*p* = 0.600
miR-199a-5p	2.420 (1.320–5.180) × 10^−2^	2.880 (1.510–5.890) × 10^−2^	*p* = 0.223
miR-210-3p	8.840 (5.680–14.300) × 10^−2^	9.150 (5.640–14.800) × 10^−2^	*p* = 0.953
miR-221-3p	0.353 (0.190–0.601)	0.340 (0.212–0.550)	*p* = 0.675
miR-342-3p	2.194 (1.409–3.206)	2.296 (1.752–3.532)	*p* = 0.341
miR-499a-5p	9.400 (3.160–17.000) × 10^−2^	14.600 (3.710–34.700) × 10^−2^	*p* = 0.010
miR-574-3p	0.108 (0.067–0.175)	0.110 (0.081–0.175)	*p* = 0.375

Data are expressed as median (25^th^ percentile–75^th^ percentile). Mann-Whitney U test was used to compare microRNA expression between women with normal and abnormal vascular endothelial function. The significance level was established at *p*-value of *p* < 0.05 (Bonferroni corrected *p*-values). Statistically significant results are marked in bold. RHI, Reactive Hyperemia Index. Higher expression rates of miR-1-3p, miR-23a-3p, and miR-499a-5p were detected in whole peripheral blood of young and middle-aged women with the occurrence of vascular endothelial dysfunction (RHI ≤ 1.67) when compared with those possessing normal vascular endothelial function (RHI > 1.67), respectively.

**Table 7 ijms-21-00430-t007:** Association between the vascular endothelial function (RHI values) and the expression of cardiovascular and cerebrovascular disease associated microRNAs in whole peripheral blood of young and middle-aged women_ overview (Spearman’s coefficient of rank correlation).

microRNA	Data Distribution RHI (Non-Normal Distribution)	Spearman’s Coefficient of Rank Correlation (rho), *p* Value
miR-1-3p	Non-normal distribution	*ρ* = −0.156, *p* = 0.012 weak negative correlation (↓ RHI ≈ ↑ miR-1-3p)
miR-16-5p	Non-normal distribution	*ρ* = −0.157, *p* = 0.361
miR-17-5p	Non-normal distribution	*ρ* = −0.083, *p* = 0.179
miR-20a-5p	Non-normal distribution	*ρ* = −0.034, *p* = 0.581
miR-20b-5p	Non-normal distribution	*ρ* = −0.060, *p* = 0.331
miR-21-5p	Non-normal distribution	*ρ* = −0.102, *p* = 0.102
miR-23a-3p	Non-normal distribution	*ρ* = −0.154, *p* = 0.013 weak negative correlation (↓ RHI ≈ ↑ miR-23a-3p)
miR-24-3p	Non-normal distribution	*ρ* = −0.055, *p* = 0.381
miR-26a-5p	Non-normal distribution	*ρ* = −0.101, *p* = 0.103
miR-29a-3p	Non-normal distribution	*ρ* = −0.060, *p* = 0.332
miR-92a-3p	Non-normal distribution	*ρ* = 0.025, *p* = 0.692
miR-100-5p	Non-normal distribution	*ρ* = −0.002, *p* = 0.981
miR-103a-3p	Non-normal distribution	*ρ* = −0.086, *p* = 0.165
miR-125b-5p	Non-normal distribution	*ρ* = −0.001, *p* = 0.987
miR-126-3p	Non-normal distribution	*ρ* = −0.047, *p* = 0.453
miR-130b-3p	Non-normal distribution	*ρ* = −0.066, *p* = 0.291
miR-133a-3p	Non-normal distribution	*ρ* = −0.054, *p* = 0.385
miR-143-3p	Non-normal distribution	*ρ* = −0.104, *p* = 0.092
miR-145-5p	Non-normal distribution	*ρ* = −0.052, *p* = 0.402
miR-146a-5p	Non-normal distribution	*ρ* = −0.022, *p* = 0.727
miR-155-5p	Non-normal distribution	*ρ* = −0.006, *p* = 0.918
miR-181a-5p	Non-normal distribution	*ρ* = −0.051, *p* = 0.411
miR-195-5p	Non-normal distribution	*ρ* = −0.073, *p* = 0.239
miR-199a-5p	Non-normal distribution	*ρ* = −0.114, *p* = 0.065
miR-210-3p	Non-normal distribution	*ρ* = −0.042, *p* = 0.505
miR-221-3p	Non-normal distribution	*ρ* = −0.041, *p* = 0.512
miR-342-3p	Non-normal distribution	*ρ* = −0.035, *p* = 0.572
miR-499a-5p	Non-normal distribution	*ρ* = −0.193, *p* = 0.002 weak negative correlation (↓ RHI ≈ ↑ miR-499a-5p)
miR-574-3p	Non-normal distribution	*ρ* = −0.102, *p* = 0.100

Correlation between variables was calculated using the Spearman’s rank correlation coefficient (*ρ*). Spearman’s rank correlation coefficient, a nonparametric measure of rank correlation, assesses how well the relationship between two variables can be described using a monotonic function. The significance level was established at a *p*-value of *p* < 0.05. Statistically significant results are marked in bold. The RHI values showed a week negative correlation with expression of miR-1-3p, miR-23a-3p, and miR-499a-5p in whole peripheral blood of young and middle-aged women. That means that women with a finding of lower RHI values showed significantly increased levels of miR-1-3p, miR-23a-3p, and miR-499a-5p in whole peripheral blood, respectively.

**Table 8 ijms-21-00430-t008:** Characteristics of women involved in the study with respect to a history of pregnancy, pregnancy-related complications and classical cardiovascular risk factors.

	Normal Pregnancies (*n* = 74)	PE (*n* = 114)	FGR (*n* = 28)	GH (*n* = 48)	*p*-Value ^1^	*p*-Value ^2^	*p*-Value ^3^
At follow-up							
Age (years) Age (range)	38.49 ± 0.40 31–50	38.05 ± 0.41 28–52	38.11 ± 0.65 32–45	38.67 ± 0.68 31–58	1.000	1.000	1.000
Time elapsed since delivery (years)	5.74 ± 0.21	5.52 ± 0.21	5.25 ± 0.35	4.96 ± 0.31	1.000	1.000	0.259
Family medical history Angina or heart attack in a first degree relative before the age of 60 years	1 (1.35%)	2 (1.75%)	0 (0%)	1 (2.08%)	-	-	-
Dispensarisation at Dpt. of Cardiology (valve problems and heart defects)	0 (0%)	1 (0.88%) Sinus tachycardia	1 (3.45%) Leaky heart valve	2 (4.17%) Mitral valve prolapse	-	-	-
On blood pressure treatment	1 (1.35%)	7 (6.14%)	0 (0%)	3 (6.25%)	-	-	-
Lipid-lowering medication	0 (0%)	0 (0%)	0 (0%)	0 (0%)	-	-	-
DM type I	0 (0%)	1 (0.88%)	0 (0%)	1 (2.08%)	-	-	-
DM type II	0 (0%)	0 (0%)	0 (0%)	0 (0%)	-	-	-
Rheumatoid arthritis	0 (0%)	0 (0%)	1 (3.45%)	2 (4.17%)	-	-	-
Chronic venous insufficiency	0 (0%)	0 (0%)	0 (0%)	1 (2.08%)	-	-	-
Thrombosis	1 (1.35%)	2 (1.75%)	1 (3.45%)	0 (0%)	-	-	-
Trombophilic gene mutations	0 (0%)	10 (8.77%)	4 (14.29%)	4 (8.33%)	-	-	-
Presence of risk factors for chronic kidney disease	0 (%)	1 (0.88%) Haematuria	0 (0%)	2 (4.17%) Abnormal kidney structure (*n* = 1) Glomerulonephritis in childhood (*n* = 1)	-	-	-
Chronic kidney disease	0 (%)	1 (0.88%) Nephrotic syndrome	0 (0%)	0 (0%)	-	-	-
**Smoking of cigarettes**							
Non-Smoker	46 (62.16%)	70 (61.40%)	22 (78.57%)	31 (64.58%)	-	-	-
Ex-smoker	17 (22.97%)	28 (24.56%)	2 (7.14%)	12 (25.0%)	-	-	-
Smoker	11 (14.86%)	16 (14.04%)	4 (14.29%)	5 (10.42%)	-	-	-
**BMI**							
Normal (<25)	56 (75.68%)	62 (54.39%)	19 (67.86%)	19 (39.58%)	-	-	-
Overweight (≥25-<30)	15 (20.27%)	32 (28.07%)	3 (10.71%)	18 (37.50%)	-	-	-
Obese (≥30)	3 (4.05%)	20 (17.54%)	6 (21.43%)	11 (22.92%)	-	-	-
**Waist circumference**							
Normal (< 80cm)	54 (72.97%)	59 (51.75%)	18 (64.29%)	15 (31.25%)	-	-	-
Central Obesity (≥80 cm)	20 (27.03%)	55 (48.25%)	10 (35.71%)	33 (68.75%)	-	-	-
**SBP**							
Normal (<120 mmHg)	56 (75.68%)	48 (42.11%)	17 (60.71%)	17 (35.42%)	-	-	-
Prehypertension (≥120–<140 mmHg)	18 (24.32%)	53 (46.49%)	11 (39.29%)	21 (43.75%)	-	-	-
Hypertension (≥140 mmHg)	0 (0%)	13 (11.40%)	0 (0%)	10 (20.83%)	-	-	-
**DBP**							
Normal (<80 mmHg)	66 (89.19%)	65 (57.01%)	18 (64.29%)	23 (47.92%)	-	-	-
Prehypertension (≥80–<90 mmHg)	6 (8.11%)	30 (26.32%)	7 (25.0%)	14 (29.16%)	-	-	-
Hypertension (≥90 mmHg)	2 (2.70%)	19 (16.67%)	3 (10.71%)	11 (22.92%)	-	-	-
**Heart rate at rest**							
Bradycardia (< 60 bpm)	7 (9.46%)	6 (5.26%)	3 (10.71%)	4 (8.33%)	-	-	-
Normal (60–100 bpm)	66 (89.19%)	108 (94.74%)	25 (89.29%)	42 (87.50%)	-	-	-
Tachycardia (>100 bpm)	1 (1.35%)	0 (0%)	0 (0%)	2 (4.17%)	-	-	-
**Serum total cholesterol**							
Normal (2.9–5.0 mmol/L)	35 (47.30%)	53 (46.49%)	15 (53.57%)	22 (45.83%)	-	-	-
High (>5.0 mmol/L)	39 (52.70%)	61 (53.51%)	13 (46.43%)	26 (54.17%)	-	-	-
**Serum HDL cholesterol**							
Normal (1.2–2.7 mmol/L)	69 (93.24%)	100 (87.72%)	24 (85.71%)	37 (77.08%)	-	-	-
Low (<1.2 mmol/L)	5 (6.76%)	14 (12.28%)	4 (14.29%)	11 (22.92%)	-	-	-
**Serum LDL cholesterol**							
Normal (1.2–3.0 mmol/L)	35 (47.30%)	38 (33.33%)	12 (42.86%)	15 (31.25%)	-	-	-
High (>3.0 mmol/L)	39 (52.70%)	76 (66.67%)	16 (57.14%)	33 (68.75%)	-	-	-
**Serum triglycerides**							
Normal (0.45–1.7 mmol/L)	73 (98.65%)	105 (92.11%)	25 (89.29%)	45 (93.75%)	-	-	-
High (>1.7 mmol/L)	1 (1.35%)	9 (7.89%)	3 (10.71%)	3 (6.25%)	-	-	-
**Serum Lp(a)**							
Normal (0–72.0 nmol/L)	63 (85.14%)	85 (74.56%)	22 (78.57%)	42 (87.50%)	-	-	-
High (>72.0 nmol/L)	11 (14.86%)	29 (25.44%)	6 (21.43%)	6 (12.50%)	-	-	-
**Serum CRP**							
Normal (0–5.0 mg/L)	69 (93.24%)	93 (81.58%)	20 (71.43%)	41 (85.42%)	-	-	-
High (>5.0 mg/L)	5 (6.76%)	21 (18.42%)	8 (28.57%)	7 (14.58%)	-	-	-
**Plasma homocysteine**							
Normal (4.4–13.6 µmol/L)	64 (86.49%)	98 (85.96%)	23 (82.14%)	44 (91.67%)	-	-	-
High (>13.6 µmol/L)	10 (13.51%)	16 (14.04%)	5 (17.86%)	4 (8.33%)	-	-	-
**Serum uric acid**							
Normal (143–339 µmol/L)	71 (95.95%)	99 (86.84%)	22 (78.57%)	41 (85.42%)	-	-	-
High (>339 µmol/L)	3 (4.05%)	15 (13.16%)	6 (21.43%)	7 (14.58%)	-	-	-
**Hormonal contraceptive use**							
No	26 (35.14%)	29 (25.44%)	7 (25.0%)	15 (31.25%)	-	-	-
In the past	25 (33.78%)	54 (47.37%)	12 (42.86%)	26 (54.17%)	-	-	-
Yes	23 (31.08%)	31 (27.19%)	9 (32.14%)	7 (14.58%)	-	-	-
**Total number of pregnancies per patient**							
1	7 (9.46%)	37 (32.46%)	8 (28.57%)	9 (18.75%)	-	-	-
2	35 (47.30%)	46 (40.35%)	13 (46.43%)	21 (43.75%)	-	-	-
3+	32 (43.24%)	31 (27.19%)	7 (25.0%)	18 (37.50%)	-	-	-
**Total parity per patient**							
1	11 (14.86%)	39 (34.21%)	11 (39.29%)	16 (33.33%)	-	-	-
2	50 (67.57%)	59 (51.75%)	15 (53.57%)	26 (54.17%)	-	-	-
3+	13 (17.57%)	16 (14.04%)	2 (7.14%)	6 (12.50%)	-	-	-
**Infertility treatment**							
Yes	3 (4.05%)	23 (20.18%)	6 (21.43%)	6 (12.50%)	-	-	-
No	71 (95.95%)	91 (79.82%)	22 (78.57%)	42 (87.50%)	-	-	-
**During gestation**							
Maternal age at delivery (years)	32.78 ± 0.38	32.26 ± 0.41	32.86 ± 0.58	33.65 ± 0.61	1.000	1.000	1.000
GA at delivery (weeks)	39.85 ± 0.10	35.91 ± 0.33	35.23 ± 0.67	38.64 ± 0.21	<0.001	<0.001	0.106
Fetal birth weight (g)	3390.14 ± 41.55	2403.45 ± 80.99	1831.43 ± 125.48	3226.46 ± 69.50	<0.001	<0.001	1.000
**Mode of delivery**							
Vaginal	69 (93.24%)	19 (16.67%)	6 (21.43%)	21 (43.75%)	<0.001	<0.001	<0.001
CS	5 (6.76%)	95 (83.33%)	23 (78.57%)	27 (56.25%)			
**Fetal sex**							
Boy	37 (50.00%)	49 (42.98%)	15 (53.57%)	23 (47.92%)	0.345	0.747	0.822
Girl	37 (50.00%)	65 (57.02%)	13 (46.43%)	25 (52.08%)			
**Blood pressure (mmHg)**							
Systolic	120.70 ± 1.13	157.32 ± 1.49	127.21 ± 3.06	148.73 ± 2.17	**<0.001**	0.232	**<0.001**
Diastolic	75.85 ± 0.76	98.71 ± 1.01	78.07 ± 2.18	94.96 ± 1.45	**<0.001**	1.000	**<0.001**

Data are presented as mean ± SE for continuous variables and as number (percent) for categorical variables. Statistically significant results are marked in bold. Continuous variables were compared using ANOVA test. Categorical variables were compared using Chi-squared test. *p*-value ^1,2,3^: the comparison among normal pregnancies and preeclampsia, fetal growth restriction or gestational hypertension, respectively. Categorical variables were compared using a chi-square test.; PE, preeclampsia; GH, gestational hypertension; FGR, fetal growth restriction; GA, gestational age; CS, Caesarean section.

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
