# Peer review of "Evaluation of Vascular Endothelial Function in Young and Middle-Aged Women with Respect to a History of Pregnancy, Pregnancy-Related Complications, Classical Cardiovascular Risk Factors, and Epigenetics"

_ijms, 2020, doi:10.3390/ijms21020430_

Round 1

Reviewer 1 Report

The manuscript is focused on evaluation of vascular endothelial function in young and middle-aged women with respect to a history of pregnancy, pregnancy-related complications, classical cardiovascular risk factors, and epigenetics.

Examined was also the association between the vascular endothelial function and expression of several microRNAs potentially involved in pathogenesis of cardiovascular/cerebrovascular diseases.

In the study are presented several interesting data. I have some questions and comments:

1. You documented higher mean expression rates of miR-1-3p, miR-23a-3p, and miR-499a-5p in whole peripheral blood of young and middle-aged women with the occurrence of vascular endothelial dysfunction when compared with women with normal vascular endothelial function.

However, there were only very weak correlations between expressions of these microRNAs and RHI values.   

Do you really mean (based also on distribution of individual samples presented in Figure 4) that screening of these particular microRNAs could be good prognostic factor and may satisfactory help to stratify a highly risky group of young and middle-aged women?

2. Do you have information how postpartum expression of microRNAs differs from expression in peripheral blood of young and middle-aged women before delivery?

3. Were the changes in miR-1-3p expression in individual peripheral blood samples followed by similar changes in expression of miR-23a-3p and miR-499a or was there variability in expression rates between mentioned microRNAs?

4. In section Biological sampling you wrote: “Fasting blood samples (from 11 hours or more) were collected at the time of the study visit.”

Please, could you describe the time of blood sample collection more in detail and could you clarify what in your case means the term “study visit”?

Author Response

Answers to reviewer 1 comments

The manuscript is focused on evaluation of vascular endothelial function in young and middle-aged women with respect to a history of pregnancy, pregnancy-related complications, classical cardiovascular risk factors, and epigenetics.

Examined was also the association between the vascular endothelial function and expression of several microRNAs potentially involved in pathogenesis of cardiovascular/cerebrovascular diseases.

In the study are presented several interesting data. I have some questions and comments:

Reviewer comment 1

You documented higher mean expression rates of miR-1-3p, miR-23a-3p, and miR-499a-5p in whole peripheral blood of young and middle-aged women with the occurrence of vascular endothelial dysfunction when compared with women with normal vascular endothelial function.

However, there were only very weak correlations between expressions of these microRNAs and RHI values.   

Do you really mean (based also on distribution of individual samples presented in Figure 4) that screening of these particular microRNAs could be good prognostic factor and may satisfactory help to stratify a highly risky group of young and middle-aged women?

Answers to reviewer comment 1

Thank you for this comment. We could see in Figure 3 that at 10.0% FPR, only a proportion of young women with the occurrence of vascular endothelial dysfunction was identified to have up-regulated expression profile of miR-1-3p (AUC 0.0601, p=0.005, sensitivity 11.76%), miR-23a-3p (AUC 0.583, p=0.024, sensitivity 17.65%), and miR-499a-5p (AUC 0.598, p=0.011, sensitivity 18.82%). Recently, we have demonstrated that epigenetic changes characteristic for cardiovascular/cerebrovascular diseases are already present in whole peripheral blood of young and middle-aged women at a higher risk of later development of cardiovascular/cerebrovascular diseases and proposed a strategy to implement screening of cardiovascular/cerebrovascular disease associated microRNAs in primary prevention programmes to identify patients at a higher risk of later development of cardiovascular/cerebrovascular diseases as soon as possible [28].

It is obvious, that vascular endothelial dysfunction is one out of multiple cardiovascular risk factors which has however only a partial impact on abnormal expression of cardiovascular and cerebrovascular disease associated microRNAs in whole peripheral blood of young and middle-aged women. That´s the reason, why we have been continuously working on the identification of all cardiovascular risk factors causing abnormal expression of cardiovascular and cerebrovascular disease associated microRNAs in whole peripheral blood of young and middle-aged women.

This information was added to the abstract, discussion and conclusion sections.

Reviewer comment 2

Do you have information how postpartum expression of microRNAs differs from expression in peripheral blood of young and middle-aged women before delivery?

Answers to reviewer comment 2

We have data on expression of cardiovascular and cerebrovascular disease associated microRNAs in whole peripheral blood of women during pregnancy, usually at the third trimester of gestation, when the pregnancy-related complications occurred and than in postpartum period within 3 to 11 years after the delivery. We plan to bring together these data later in another manuscript. But at this moment, we can say that miR-1-3p and miR-23a-3p expression levels did not show any changes in whole peripheral blood of women during gestation when patients with and without pregnancy-related complications were compared. MiR-499a-5p expression levels were significantly decreased, but just in patients with the occurrence of gestational hypertension when compared with normal pregnancies. On the other hand, miR-1-3p postpartum expression levels were significantly increased in women, who suffered from gestational hypertension or preeclampsia. MiR-499a-5p postpartum expression levels were significantly increased in women, who developed gestational hypertension. MiR-23a-3p postpartum expression levels showed no changes in whole peripheral blood of women when patients with and without a history of pregnancy-related complications were compared.

This information was added to the discussion section.

Reviewer comment 3

Were the changes in miR-1-3p expression in individual peripheral blood samples followed by similar changes in expression of miR-23a-3p and miR-499a or was there variability in expression rates between mentioned microRNAs?

Answers to reviewer comment 3

Thank you for this comment. We checked this point and found out that the dysregulation of these 3 microRNAs was mutually independent. All these 3 microRNAs were simultaneously dysregulated just in 4 patients with abnormal RHI values. MiR-1-3p dysregulation was followed by miR-23a-3p dysregulation just in 1 patient with abnormal RHI values. MiR-1-3p dysregulation was followed by miR-499a-5p dysregulation also just in 1 patient with abnormal RHI values.

This information was added to the results section.

Reviewer comment 4

In section Biological sampling you wrote: “Fasting blood samples (from 11 hours or more) were collected at the time of the study visit.”

Please, could you describe the time of blood sample collection more in detail and could you clarify what in your case means the term “study visit”?

Answers to reviewer comment 4

4.5. Biological sampling

For detailed information please see our previous publication [26]. Fasting blood samples were collected at the time of the study visit, 3 to 11 years postpartum, when comprehensive examination of patient was performed (blood pressure measurements, BMI and waist circumference measurements). Patients were told not to consume anything but water for 12 hours leading up to the test (collection of whole peripheral blood samples was performed within 9.00 a.m. to 10.00 a.m. GMT). Total serum cholesterol, high-density lipoprotein (HDL) cholesterol, low-density lipoprotein (LDL) cholesterol, triglycerides, lipoprotein A Lp(a), high-sensitivity C-reactive protein (CRP), uric acid, and plasma homocysteine were analysed using standard laboratory methods at the Institute for the Care of the Mother and Child.

This information was added to the methods section.

Reviewer 2 Report

This is a nice study that aimed at investigating the effect of previous pregnancies and classical cardiovascular risk factors on vascular endothelial function in a group of young and middle-aged women 3 to 11 years postpartum. The applied methods included the examination of a large set of factors regarding microvascular functions, patient history, clinical characteristics and peripheral blood laboratory parameters including expression of microRNAs involved in pathogenesis of cardiovascular/cerebrovascular diseases. The authors found that a proportion of overweight women and women with central obesity had lower Reactive Hyperemia Index (RHI) values, and that a proportion of women with vascular endothelial dysfunction had up-regulated expression of three microRNAs in whole peripheral blood. The expression of these microRNAs was negatively correlated with RHI values. The authors concluded that the screening of these microRNAs associated with vascular endothelial dysfunction may help to stratify a highly risky group of young and middle-aged women that would benefit from early implementation of primary prevention strategies.

Overall, this is a very nice study and paper. The study design and writing of the paper is meticulous, the presentation of the large set of data is clear, the results are interesting, and the introduction and discussion are very comprehensive. Therefore, I suggest to accept the paper for publication after the following minor issues are addressed:

There was a difference in RHI values regarding the presence or absence of central obesity and according to BMI intervals. It would be interesting to see whether there is also a difference in microRNA expression according to these sub-categories. Negative results are discussed more extensively than positive results. I would expand the paragraph on microRNAs and also describe the biological functions and clinical significance of the three microRNAs that were found to be changed. Do these have any relation to endothelial functions? Where are they produced? Were these previously found to be changed in expression in complications of pregnancy? When?

Author Response

Answers to reviewer 2 comments

This is a nice study that aimed at investigating the effect of previous pregnancies and classical cardiovascular risk factors on vascular endothelial function in a group of young and middle-aged women 3 to 11 years postpartum. The applied methods included the examination of a large set of factors regarding microvascular functions, patient history, clinical characteristics and peripheral blood laboratory parameters including expression of microRNAs involved in pathogenesis of cardiovascular/cerebrovascular diseases. The authors found that a proportion of overweight women and women with central obesity had lower Reactive Hyperemia Index (RHI) values, and that a proportion of women with vascular endothelial dysfunction had up-regulated expression of three microRNAs in whole peripheral blood. The expression of these microRNAs was negatively correlated with RHI values. The authors concluded that the screening of these microRNAs associated with vascular endothelial dysfunction may help to stratify a highly risky group of young and middle-aged women that would benefit from early implementation of primary prevention strategies.

Reviewer´s 2 comments

Overall, this is a very nice study and paper. The study design and writing of the paper is meticulous, the presentation of the large set of data is clear, the results are interesting, and the introduction and discussion are very comprehensive. Therefore, I suggest to accept the paper for publication after the following minor issues are addressed:

There was a difference in RHI values regarding the presence or absence of central obesity and according to BMI intervals. It would be interesting to see whether there is also a difference in microRNA expression according to these sub-categories.

Answers to reviewer´s 2 comments

No association was found between expression of the relevant cardiovascular and cerebrovascular-disease associated microRNAs (miR-1-3p: p=0.848, miR-23a-3p: p=0.962 and miR-499a-5p: p=0.473) and the presence of central obesity. Similarly, no association was observed between expression of the relevant cardiovascular and cerebrovascular-disease associated microRNAs (miR-1-3p: p=0.367, miR-23a-3p: p=0.902 and miR-499a-5p: p=0.264) and the presence of overweight/obesity.

Since no association was identified, this information was not added to the manuscript.

Reviewer´s 2 comments

Negative results are discussed more extensively than positive results. I would expand the paragraph on microRNAs and also describe the biological functions and clinical significance of the three microRNAs that were found to be changed. Do these have any relation to endothelial functions? Where are they produced? Were these previously found to be changed in expression in complications of pregnancy? When?

Answers to reviewer´s 2 comments

The required information was added to the discussion section.

Discussion

MiR-1-3p is generated from miR-1-1 and miR-1-2 precursors whose genes are located on chromosomes 20q13.3 and 18q11.2. MiR-1 is highly expressed in heart muscle, especially in the myocardium, and in skeletal muscles [80-82].  Circulating miR-1 levels are significantly elevated in acute myocardial infarction and correlate with circulating troponin T levels, a marker of cardiac damage [83]. MiR-1 is a potential target of therapeutic intervention in cardiovascular diseases, cardiac ischemia and complications following myocardial infarction. Inhibition of miR-1 by oligonucleotides is cardioprotective, as it leads to a reduction in apoptosis, an increase in resistance to oxidative stress and a reduction in spontaneous arrhythmias [84-86]. In addition, SUR2B/Kir6.1 channel openers correct endothelial dysfunction in chronic heart failure via the miR-1-3p/ET-1 pathway [87]. We recently reported that a proportion of women with up-regulated postpartal miR-1-3p expression profile in whole peripheral venous blood, who had a history of gestational hypertension or preeclampsia in their previous pregnancies, had an increased cardiovascular risk and should be monitored long-term [28]. Parallel, up-regulated miR-1-3p profile was observed in placental tissue and umbilical cord blood samples in women with the onset of preeclampsia and FGR [88, 89]. But miR-1-3p expression levels did not show any changes in whole peripheral blood of women during gestation when patients with and without pregnancy-related complications were compared [90].

MiR-23a, encoded by a gene located at chromosome 19p13.12, forms two mature microRNAs: miR-23a-5p and miR-23a-3p. MiR-23a regulates cardiomyocyte apoptosis, a key pathogenesis factor of heart failure, by targeting manganese superoxide dismutase gene [91]. MiR-23a also regulates the vasculogenesis of coronary artery disease via targeting epidermal growth factor receptor [92]. Circulating miR-23a may be a new biomarker for coronary artery disease, since increased levels of miR-23a predict the presence and severity of coronary lesions in patients with coronary artery disease [92]. MiR-23a-3p also suppressed oxidative stress injury in a mouse model of focal cerebral ischemia-reperfusion [93]. Furthermore, miR-23c, but not miR-23a was reported to function as a new regulator to inhibit angiogenesis by targeting SDF-1α, which is responsible for delayed process of wound healing in patients with diabetic foot ulcer due to reduced expression of growth factors, persistant inflammation and endothelial dysfunction [94]. Since our recent study demonstrated upregulation of miR-23a-3p in a proportion of children with normal clinical findings born of gestational hypertension complicated pregnancies only, we suppose that compensatory effect of miR-23a-3p may appear more likely in these children to normalise cardiomyocyte state and vasculogenesis [95]. No change in miR-23a-3p expression profile was observed in various biological samples (whole peripheral venous blood, placental tissue, umbilical cord blood) when women with pregnancy-related complications and normal pregnancies were compared both during gestation or postpartum periods [88-90].

MiR-499a-5p, encoded by the miR-499a gene located on chromosome 20q11.22, is strongly expressed in the heart under physiological conditions. MiR-499 is involved in inhibiting cardiomyocyte apoptosis by suppressing calcineurin-mediated dephosphorylation of the Drp1 protein. It has also been reported that p53 downregulates the expression of miR-499 [96]. MiR-499 is also associated significantly with myocardial infarction, and together with miR-133a and miR-208a, they represent potential early diagnostic biomarkers of myocardial infarction [97-100]. Furthermore, miR-499 is significantly involved in the inflammatory signaling pathways of bronchial asthma [101]. No data on the role of miR-499a-5p in pathogenesis of endothelial dysfunction are currently available. Our recent study demonstrated upregulation of miR-499a-5p in a proportion of women with a history of gestational hypertension in previous pregnancy. Considering the clinical relevance of miR-499, we concluded that the group of women with the upregulated miR-499a-5p expression profile represented a high-risk group of patients who would benefit from early implementation of prevention programs and long-term monitoring [28]. Placental tissue demonstrated up-regulated miR-499a-5p expression profile in women with the occurence of gestational hypertension, preeclampsia and FGR [88]. Nevertheless, in whole periheral venous blood of women with the onset of gestational hypertension miR-499a-5p levels were significantly decreased [90].